# A Characterization Theorem for Equivariant Networks with Point-wise Activations

**Marco Pacini**
Fondazione Bruno Kessler, Italy
mpacini@fbk.eu

**Xiaowen Dong**
University of Oxford, United Kingdom
xdong@robots.ox.ac.uk

**Bruno Lepri**
Fondazione Bruno Kessler, Italy
lepri@fbk.eu

**Gabriele Santin**
University of Venice, Italy
gabriele.santin@unive.it

## Abstract

Equivariant neural networks have shown improved performance, expressiveness and sample complexity on symmetrical domains. But for some specific symmetries, representations, and choice of coordinates, the most common point-wise activations, such as ReLU, are not equivariant, hence they cannot be employed in the design of equivariant neural networks. The theorem we present in this paper describes all possible combinations of finite-dimensional representations, choice of coordinates and point-wise activations to obtain an exactly equivariant layer, generalizing and strengthening existing characterizations. Notable cases of practical relevance are discussed as corollaries. Indeed, we prove that rotation-equivariant networks can only be invariant, as it happens for any network which is equivariant with respect to connected compact groups. Then, we discuss implications of our findings when applied to important instances of exactly equivariant networks. First, we completely characterize permutation equivariant networks such as Invariant Graph Networks with point-wise nonlinearities and their geometric counterparts, highlighting a plethora of models whose expressive power and performance are still unknown. Second, we show that feature spaces of disentangled steerable convolutional neural networks are trivial representations.

## 1 Introduction

In recent years, equivariant neural networks have improved the performance of standard neural networks by harnessing symmetries of the training data, and have risen to an entirely new branch of machine learning known as geometric deep learning (Bronstein et al., 2021; 2017; Kondor & Trivedi, 2018). Those networks (Wood & Shawe-Taylor, 1996; Cohen & Welling, 2016a) have shown improved generalization capabilities across various research areas, including computer vision (Worrall et al., 2017; Hoogeboom et al., 2018; Dieleman et al., 2016), computer graphics (Weiler et al., 2018; Zaheer et al., 2017; Qi et al., 2017; Maron et al., 2020; Marcos et al., 2017; Sifre & Mallat, 2013), and graph learning (Maron et al., 2018; 2019a;b; Geerts, 2020; Kondor et al., 2018; Hy et al., 2018). However, these networks pose additional constraints in the architecture design. Namely, the use of pointwise activations, which are fundamental in common neural networks for their simplicity, easiness to train, and computational efficiency, is limited by a lack of equivariance in certain settings (Cohen & Welling, 2016b; Weiler et al., 2018). Their application in equivariant models is thus restricted and needs a careful choice, which is however only partially explored. Indeed, pioneering work (Wood & Shawe-Taylor, 1996) proved a theorem that enables researchers, having at their disposal certain representations, to be automatically informed of all employable point-wise activations and vice-versa. This, to the best of our knowledge, is the first and only characterization theorem for equivariant networks with point-wise activations, which has however significant limitations in several aspects, namely, (i) it only applies to finite groups, (ii) it is redundant, since its classification is unable to identify representations that are equivalent up to isomorphisms or change of bases, and (iii) it is restricted to a subclass of possible activation functions.

In this paper, using tools from representation theory (Fulton & Harris, 2004) and matrix group theory (Flor, 1969), we present a stronger and more general theorem that fills the described gaps in the result of Wood & Shawe-Taylor (1996). In more details, we consider classes $\mathcal{F}$ of activation functions and groups $\mathcal{M}$ of representation matrices, and we investigate when they can be combined to obtain an equivariant layer, i.e., they commute–see Section 4. We start by defining operations that map any $\mathcal{F}$ to a corresponding maximal admissible group of matrices $\mathcal{M}(\mathcal{F})$, and vice-versa map $\mathcal{M}$ to a maximal admissible family of activations $\mathcal{F}(\mathcal{M})$. Crucially, we highlight the dual nature of those operations and how their composition stabilizes, leading to a finite family of explicitly defined maximal classes. Finally, for these few maximal classes we exhibit the dual pairings $(\mathcal{F}, \mathcal{M})$, thus obtaining a complete and exhaustive categorization of admissible activation-representation pairs, which includes the classification of Wood & Shawe-Taylor (1996) as a special case.

We then apply this theorem to obtain a number of novel results. First, we leverage it to collect new theoretical insights into existing methods. Namely, we prove a stronger result than Wood & Shawe-Taylor (1996) in the case of finite groups (Section 5.1) by identifying isomorphic admissible representations. Moreover, we consider disentangled equivariant networks (Cohen & Welling, 2016b), and we show that they admit point-wise activations if and only if the linear representations underlying their feature spaces are trivial (Section 5.2). Second, we specialize our results to particular cases of equivariant networks of practical relevance (Section 6). For rotation-equivariant networks (Weiler et al., 2018), we show that all admissible networks are invariant, and hence unable to learn many equivariant tasks such as segmentation or detection. This proves a significant barrier for the use of equivariant networks with point-wise activations in this kind of tasks. Instead, for Invariant Graph Networks (IGNs) (Maron et al., 2018) and geometric IGNs (Dym & Gortler, 2022; Finkelshtein et al., 2022; Joshi et al., 2023), including the $k$-order invariant and equivariant networks presented by Maron et al. (2018), we show that the choice of admissible representation layers compatible with point-wise activations is not only limited to those described by Maron et al. (2018), and in fact we highlight and fully characterize a wider class of permutation equivariant networks in terms of subgroups of the symmetric group. This opens new directions and provides novel tools for the design of equivariant networks beyond the known ones.

We point out that our approach has limitations, since we focus solely on *exact* equivariance with respect to arbitrary *finite-dimensional* representations. Those representations encompass disentangled representations as well as regular representations (Cohen & Welling, 2016a;b), and thus we address a substantial portion of documented use cases in the literature. Nevertheless, our work does not apply to infinite-dimensional representations, and in particular to the vast body of literature utilizing harmonic analysis techniques. Also in this case, however, our results have a space of application. Indeed, for computational reasons these infinite-dimensional models are often transformed into finite ones in diverse ways, and when this is achieved through the discretization of both the symmetry group and the domain, the resulting discrete model can be frequently analyzed using our approach. Instead, we can not analyze those cases where the discretized model does not exhibit exact equivariance with respect to the initial symmetry group, even if for these models the empirical evidence suggests that this equivariance is approximately maintained (Cohen et al., 2018). Moreover, it is crucial to emphasize that further alternative approaches exist in the literature, and unexplored possibilities remain.

In brief, our contributions are summarized as follows: (i) we provide a characterization theorem for equivariant neural networks with continuous point-wise activations, by showing the existence of a finite number of maximal sets of equivariant classes, enumerating them, and providing an explicit dual pairing between activations and representations, (ii) we use this result to give an exhaustive description of networks that are equivariant with respect to finite groups, and to show a barrier in the use of disentangled equivariant networks, and (iii) we discuss implications of this theorem in practical and relevant scenarios, namely highlighting a severe limitation of rotation-equivariant networks, while providing new and unexplored design possibilities for (geometric) IGNs.

The paper is organized as follows: Section 2 provides an overview of related work on equivariant models and existing limitations concerning point-wise activations. Section 3 provides preliminaries for our work. In Section 4, we present the general formulation of the characterization theorem for equivariant networks with point-wise activations. Section 5 explores significant implications of the theorem for specific but still abstract cases such as finite groups or disentangled networks. In Section 6 we discuss examples of relevant significance in practical scenarios such as equivariance with respect to the symmetric group and the rotation group and their application to graph invari-

ant networks, networks equivariant with respect to Euclidean transformations, and geometric graph processing. Finally, Section 7 summarizes our findings and discusses future research directions.

## 2 RELATED WORK

Historically, early explicit integration of representation theory and harmonic analysis into machine learning can be dated to Kakarala (1992) and Kondor (2008). Wood & Shawe-Taylor (1996) are the first to bring equivariance into deep learning with a general approach. They define equivariant neural networks and give a classification of those models for the case of point-wise activations. In more recent years, Cohen & Welling (2014; 2016a;b) presented the foundational work of group equivariant convolutional networks and introduced the general model of steerable Convolutional Neural Networks (CNNs) (Cohen & Welling, 2016b), a popular and efficient class of equivariant models. In the following years many equivariant models and many applications to different domains appeared in the literature: rotation-invariance for galaxy morphology prediction (Dieleman et al., 2015), permutation invariance for set processing (Qi et al., 2017; Zaheer et al., 2017), permutation invariance for graph and relational structure learning (Maron et al., 2018; Kondor et al., 2018; Pan & Kondor, 2022), simultaneous roto-translation invariance and permutation-invariance for 3D point-cloud processing (Thomas et al., 2018; Dym & Maron, 2020), and roto-translation invariance for medical image analysis (Bekkers et al., 2018). Different research directions focused instead on theoretical aspects of equivariant models, including the design of new frameworks (Cohen et al., 2019), proofs of expressivity and universality (Maron et al., 2019b; Geerts, 2020; Ravanbakhsh, 2020; Zhou, 2020; Yarotsky, 2018; Joshi et al., 2023), generalization bounds and sample complexity (Behboodi et al., 2022; Cohen et al., 2019; Sannai et al., 2021; Zweig & Bruna, 2021; Elesedy, 2022), and characterizations (Wood & Shawe-Taylor, 1996; Kondor & Trivedi, 2018; Lang & Weiler, 2021). Despite this extensive theoretical investigation, a comprehensive study of the interaction of activations and representations is still missing. This constitutes a potential limitation in the understanding and design of novel equivariant architectures enjoying certain desired properties. This paper aims at filling this gap. We remark finally that there exist activations which are not point-wise, and that they are sometimes employed in practice even if they are less computationally favourable, and their investigation is still at its infancy. Indeed, to the best of our knowledge, a general framework to describe and study them in a principled manner is yet to be proposed. Examples of those activations are the norm nonlinearities (Worrall et al., 2017), squashing nonlinearities (Sabour et al., 2017), tensor product nonlinearities (Kondor, 2018), and gated nonlinearities (Weiler et al., 2018). We refer to Weiler & Cesa (2019) for further details.

## 3 PRELIMINARIES

### 3.1 GROUPS, REPRESENTATIONS AND AFFINE TRANSFORMATIONS

We would like to define functions symmetrical with respect to a certain set of transformations. Classes of transformations suitable for computation and technical manipulation are groups (Fulton & Harris, 2004). A group is a set of elements which can be composed together, can be inverted and such that there exists an element neutral with respect to composition. For further details we refer to Definition 6 and Example 2 in Appendix A.2.

Despite the properties presented by groups, these algebraic structures need adaptation to work with the language of linear algebra typical of neural networks. The right tool to operate this translation is representation theory (Fulton & Harris, 2004) which studies how abstract groups can be translated to sets of matrices which are group themselves. Given a group $G$, a vector space $V$ on the field $\mathbb{R}$ of real numbers, and the set $\mathrm{GL}(V)$ of linear invertible functions from $V$ to itself, a $G$-representation is a function $\rho : G \to \mathrm{GL}(V)$ compatible with the group structures. When possible we will indicate such a representation by using simply $V$ (Definition 10). We will write $\mathrm{GL}_n(\mathbb{R})$ when $V = \mathbb{R}^n$.

For our purposes some particular representations will play an important role. Given an action of $G$ on a finite set $X$ with cardinality $n$, setting $V = \mathbb{R}^X$, and considering the standard basis $\mathcal{B} = \{e_i\}_{i=1}^n$ of $X$, a *permutation representation* is a representation such that $g(e_i) = e_{gi}$ for each $g \in G$ and $i \in X$, and a *signed permutation representation* is such that $g(e_i) = \pm e_{gi}$. Moreover, a *monomial representation* is such that $g(e_i) = a_{g,i} e_{gi}$ for some non-null real number $a_{g,i}$, and if additionally

$a_{g,i}$ is positive we say that the monomial representation is *non-negative*. If each $a_{g,i}$ belongs to the multiplicative group $\langle b^n \rangle_{n \in \mathbb{Z}}$ for a real value $0 < b \neq 1$, we say that the representation is *b-monomial*. Similarly if each $a_{g,i}$ belongs to $\langle \pm b^n \rangle_{n \in \mathbb{Z}}$, we say that the representation is $\pm b$-*monomial* (Definition 12). These representations respectively generate the group of permutation matrices, signed-permutation matrices, monomial matrices, non-negatove monomial matrices, and $b$- and $\pm b$-monomial matrices in $\text{GL}_n(\mathbb{R})$. The following matrices $P, S$ and $M$ are respectively examples of permutation matrices, signed permutation matrices and 2-monomial matrices:

$$P = \begin{bmatrix} 0 & 0 & 1 \\ 1 & 0 & 0 \\ 0 & 1 & 0 \end{bmatrix}, \quad S = \begin{bmatrix} 0 & 0 & 1 \\ 1 & 0 & 0 \\ 0 & -1 & 0 \end{bmatrix}, \quad M = \begin{bmatrix} 0 & -\frac{1}{2} & 0 \\ 0 & 0 & 2 \\ 2 & 0 & 0 \end{bmatrix}.$$

If $V$ and $W$ are vector spaces underlying representations $\rho_V$ and $\rho_W$ of the same group $G$, we say that a function $f : V \rightarrow W$ is $G$-equivariant if $f \circ \rho_V = \rho_W \circ f$. We write $\text{Hom}(V, W)$ for the set of all linear maps from $V$ to $W$, and define $\text{Hom}_G(V, W)$ as the set of $G$-equivariant linear functions from $V$ to $W$. For the interest of this work, we consider affine maps between $V$ and $W$, i.e., a composition between a linear map $\text{Hom}(V, W)$ and a translation on $W$. We denote as $\text{Aff}(V, W)$ those maps and as $\text{Aff}_G(V, W)$ the set of equivariant affine functions (see also Appendix A.5).

## 3.2 EQUIVARIANT NEURAL NETWORKS

We now define equivariant neural networks, which were first introduced by Wood & Shawe-Taylor (1996) as *Group Representation Networks* and by Cohen & Welling (2016b) as *G-Steerable Convolutional Networks*. The same model later appears in other papers such as Behboodi et al. (2022) under the name of *equivariant neural networks*, which is the notation we adopt in this paper.

Given a group $G$ and arbitrary $G$-representations $V_i$ for $1 \leq i \leq m$, a $G$-equivariant neural network is a composition

$$\Phi = \phi_m \circ \tilde{f}_{m-1} \circ \phi_{m-1} \circ \cdots \circ \tilde{f}_1 \circ \phi_0, \tag{1}$$

where each *activation* $\tilde{f}_i : V_i \rightarrow V_i$ is a $G$-equivariant function, and $\phi_i \in \text{Aff}_G(V_i, V_{i+1})$ is an affine $G$-equivariant map.

An activation $\tilde{f} : V_i \rightarrow V_i$ is *point-wise* if there exist a basis $\mathcal{B}_i = \{v_1, \ldots, v_m\}$ of $V_i$ and a real scalar function $f$ such that

$$\tilde{f}(a_1 v_1 + \cdots + a_m v_m) = f(a_1)v_1 + \cdots + f(a_m)v_m \quad \forall a_1, \ldots, a_m \in \mathbb{R}.$$

In this case, we say that $f$ induces $\tilde{f}$ on $\mathcal{B}$. Note that we do not constrain activations to be non-linear or non-affine but, from now on, we only consider continuous functions $\mathcal{C}(\mathbb{R})$. This condition is not particularly restrictive as continuous functions constitute a wide class of function which includes all commonly employed point-wise activations (such as ReLU, $\tanh$, ...), they are compatible with backpropagation, and strictly encompasses activations dealt by Wood & Shawe-Taylor (1996).

We remark that other works (Kondor & Trivedi, 2018; Cohen et al., 2019) have considered different definitions of equivariant networks, which allow one to treat infinite dimensional representations, but are limited to represent signals defined on homogeneous spaces. As mentioned in Section 1, by avoiding these infinite dimensional representations we fail to fully describe various physical phenomena that are modeled as continuous, and therefore belonging to infinite-dimensional spaces, which are however discretized and often reduced to finite models in the computational practice.

## 4 MAIN RESULT

We present a stronger and more general version of the characterization theorem proposed in Wood & Shawe-Taylor (1996), which is the only existing partial characterization of admissible activation functions to the best of our knowledge. Although building on the existing results of Wood & Shawe-Taylor (1996), we introduce several extensions and improvements, namely i) we generalize the theorem to non-finite groups and continuous activations, such as non-discrete ones as rotations and Euclidean transformations (Weiler et al., 2018); ii) We identify certain network classes by considering representations up to isomorphism; iii) We integrate the theory with several adjustments that make the classification effective, such as the maximality discussed in Lemma 1; iv) We ground

our characterization on the classification of multiplicative subgroups of $\mathbb{R}$, which highlights the potential to generalize these results to different scenarios. For the reader's convenience, we present the general result in Theorem 1, and its specialization to compact groups in Theorem 2.

For our purposes, we consider the following sets $\mathcal{F}$ of functions as activation functions:

- Continuous functions, $\mathcal{C}(\mathbb{R})$,
- The set of $b$-multiplicative functions for a given real value $b > 1$, namely each $f \in \mathcal{C}(\mathbb{R})$ such that $f(b^n x) = b^n f(x)$ for each $n \in \mathbb{Z}$ and for each $x \in \mathbb{R}$ (see Lemma 8 in Appendix A.8 for a more explicit definition),
- The set of $\pm b$-multiplicative functions for a given real value $b > 1$, namely each $f \in \mathcal{C}(\mathbb{R})$ such that $f(\pm b^n x) = \pm b^n f(x)$ for each $n \in \mathbb{Z}$ and for each $x \in \mathbb{R}$ (see Lemma 8 in Appendix A.8 for a more explicit definition),
- Odd functions, namely each $f \in \mathcal{C}(\mathbb{R})$ such that $f(-x) = -f(x)$ for each $x \in \mathbb{R}$,
- Semilinear functions, namely each $f \in \mathcal{C}(\mathbb{R})$ that is linear on $\mathbb{R}_{>0}$ and on $\mathbb{R}_{<0}$.

For our construction we will need also the following relations between activations and matrices. Given a group $G$ and a representation $\rho : G \to \mathrm{GL}(V)$ of $G$, we consider a point-wise activation $\tilde{f} : V \to V$ induced by $f : \mathbb{R} \to \mathbb{R}$ on a basis $\mathcal{B}$ of $V$. The image of $\rho$ in $\mathrm{GL}(V)$ with respect to the basis $\mathcal{B}$ forms a group of matrices which we denote $\mathcal{M}$, and note that $\tilde{f}$ is $G$-equivariant if and only if it commutes with respect to the matrices in $\mathcal{M}$, i.e., $\tilde{f}(Mv) = M\tilde{f}(v)$ for each $M \in \mathcal{M}$ and $v \in V$, where both $v$ and $M$ are written on the basis $\mathcal{B}$.

This paper proposes a simple procedure to recover the maximal group of representation matrices $\mathcal{M}$ compatible with a given class $\mathcal{F}$ of activation functions, and vice-versa the widest class of functions $\mathcal{F}$ given a group of representation matrices $\mathcal{M}$. The precise definition is as follows.

**Definition 1.** *Given $\mathcal{F} \subseteq \mathcal{C}(\mathbb{R})$, we define the maximal group $\mathcal{M}(\mathcal{F})$ admitted by $\mathcal{F}$ as the set of all matrices in $\mathrm{GL}_n(\mathbb{R})$ which commutes with each $\tilde{f}$ induced by $f \in \mathcal{F}$. Conversely, given $\mathcal{M} \subseteq \mathrm{GL}_n(\mathbb{R})$, the maximal set $\mathcal{F}(\mathcal{M})$ admitted by $\mathcal{M}$ is the set of all functions $f \in \mathcal{C}(\mathbb{R})$ such that $\tilde{f}$ induced by $f$ commutes with each $M \in \mathcal{M}$.*

Observe that $\mathcal{M}(\mathcal{F})$ has trivially a group structure with respect to matrix product, since if $M_1, M_2 \in \mathcal{M}(\mathcal{F})$ commute with $\tilde{f}$, then also their product $M_1 M_2$ does. Moreover, the following stabilization lemma (Lemma 1) proves that the maps to the maximal sets (Definition 1) stabilize after two iterations for any initial choice of $\mathcal{M}$ or $\mathcal{F}$, proving the duality of the operators $\mathcal{M}(\cdot)$ and $\mathcal{F}(\cdot)$ on maximal elements. See Section A.8 for the proof.

**Lemma 1.** *The group of matrices $\mathcal{M}' = \mathcal{M}(\mathcal{F}(\mathcal{M}))$ is the largest group in $\mathrm{GL}_n(\mathbb{R})$ for which $\mathcal{F}(\mathcal{M}') = \mathcal{F}(\mathcal{M})$, and $\mathcal{F}' = \mathcal{F}(\mathcal{M}(\mathcal{F}))$ is the largest family of functions in $\mathcal{C}(\mathbb{R})$ for which $\mathcal{M}(\mathcal{F}') = \mathcal{M}(\mathcal{F})$.*

Thanks to Lemma 1, it is sufficient to provide explicit pairings between $\mathcal{M}$ and $\mathcal{F}(\mathcal{M})$ (or $\mathcal{F}$ and $\mathcal{M}(\mathcal{F})$) just for maximal admissible groups $\mathcal{M}$, and similarly for maximal admissible families of functions $\mathcal{F}$. Indeed, given any other $\overline{\mathcal{M}}$ (or $\overline{\mathcal{F}}$) which is not maximal, it is sufficient to find a superset $\mathcal{M}$ of $\overline{\mathcal{M}}$ which is a maximal group in the sense of Definition 1, and this will give a set of admissible functions $\mathcal{F}$ which are equivariant also for $\overline{\mathcal{M}}$.

We can now state our main result, which shows that, under mild assumptions, these maximal groups (or, equivalently, maximal function classes) are only in a very limited number. Moreover, for each of these we provide the exact correspondence between $\mathcal{M}$ and $\mathcal{F}$, thus providing an exhaustive classification of all possible admissible pairs of $\mathcal{M}$ and $\mathcal{F}$. As the set $\mathcal{F}$ represents the activation functions, we make the basic assumptions that it does not contain only affine functions. We remark that the proof of this theorem, for which we refer to Section A.8, uses a novel algebraic approach that makes it easier to generalize the result to other scenarios, as discussed in Section 7.

**Theorem 1.** *Assume that $\mathcal{F}$ is not a set of affine functions. The following are exactly all dual pairs of maximal admissible families of activations $\mathcal{F}$ and associated maximal admissible groups $\mathcal{M}$:*

*1. Continuous functions and permutation matrices,*

2. *Odd continuous functions and signed permutation matrices,*

3. *Semilinear functions and non-negative monomial matrices,*

4. *Continuous $b$-multiplicative functions and $b$-monomial matrices,*

5. *Continuous $\pm b$-multiplicative functions and $\pm b$-monomial matrices.*

Note that some groups of matrices or families of functions are contained in each other. For example, permutation matrices are signed permutation matrices, while $b^2$-monomial matrices are contained into $b$-monomial matrices, but they are still maximal following Definition 1.

We classified groups of matrices and families of activation functions commuting with each others, and this classification is exhaustive. However, for compact groups $G$ we show that there is another tool to enlarge the set of possible activation functions, namely moving to isomorphic representation which admit a wider family of activation functions. This approach solves and generalizes an issue raised in Wood & Shawe-Taylor (1996), where in Section 4.2 it is noted that the representation $\rho : \mathbb{Z}_2 \to \mathrm{GL}_2(\mathbb{R})$ given by

$$\rho(0) = \begin{bmatrix} 1 & 0 \\ 0 & 1 \end{bmatrix}, \quad \rho(1) = \begin{bmatrix} 0 & 2 \\ \frac{1}{2} & 0 \end{bmatrix}$$

only admits semi-linear activation functions, according to the rather narrow set of activations studied by Wood & Shawe-Taylor (1996). To be more precise, by expanding the class of activation functions to continuous functions, we have shown in Theorem 1 that in fact the broader set of 2-multiplicative functions is admissible. Despite 2-multiplicative functions being a larger set than semi-linear functions, they still do not include common activations such as $\tanh$, sigmoid, or softplus. However, by a basis change through the matrix $B = \mathrm{diag}(1/\sqrt{2}, \sqrt{2})$, we obtain the isomorphic representation

$$B\rho(0)B^{-1} = \begin{bmatrix} 1 & 0 \\ 0 & 1 \end{bmatrix}, \quad B\rho(1)B^{-1} = \begin{bmatrix} 0 & 1 \\ 1 & 0 \end{bmatrix},$$

which is the standard permutation representation of $\mathbb{Z}_2$, and thus commutes with all continuous functions (see Theorem 1). This approach is generalized to the case of arbitrary compact groups by the following theorem, showing that in this case representations are always isomorphic to (signed) permutation representations, which have the widest admissible family of activations. We refer to Section A.8 for its proof.

**Theorem 2.** *Let $G$ be a compact group and assume that $\mathcal{F}$ is not a set of affine functions. Then any representation of $G$ admitted by $\mathcal{F}$ is isomorphic to a subgroup of the (signed) permutation matrices. Thus, $\mathcal{F}$ can always be chosen as the set of (odd) continuous functions.*

## 5 Relevant Implications for Practical Scenarios

In this section, we delve into the practical implications of Theorems 1 and 2, focusing on non-odd non-affine activations, the most used activations in practice. We further specialize these results to the case of finite groups, where the only non-trivial admissible representations is induced by permutation representations, which can be described with more precision. We explore the implications of these results on neural networks designed for processing first-order relational structures like sets and point clouds (Qi et al., 2017; Zaheer et al., 2017). These networks have proven highly effective in practice, showcasing efficiency and accuracy when dealing with such unordered data. Finally, in this setting, we show that admissible disentangled representations coupled with point-wise activations are trivial.

### 5.1 Finite Groups

As the majority of activations used in practice are induced by non-odd non-linear functions, we present a corollary of Theorem 2 that completely describes representations for finite groups that can be used in these cases. We show that admissible representations are only permutations representations up to positive-scaling of the basis.

**Corollary 1.** *Let $G$ be a finite group, and let $f : V \to V$ be a non-odd non-affine equivariant activation function on a $G$-representation $V$ defined on the basis $\mathcal{B}$. Then there exists a positive scaling of $\mathcal{B}$ and a collection of finite index subgroups $H_i < G$ such that $V = \mathbb{R}^{G/H_1} \times \cdots \times \mathbb{R}^{G/H_m}$.*

*Proof.* We can consider $V$ to be a finite permutation representation thanks to Theorem 2, as the only admissible bounded matrix groups are permutation matrices, while signed permutation matrices are compatible only with odd functions. A permutation representation has an underlying permutation set that can be decomposed in the disjoint union of orbits. Recall that, for any $X$ on which $G$ acts transitively, there exists a set bijection $X \cong G/H$ for some subgroup $H$ of $G$. Hence, a permutation representation $V$ can be decomposed into the direct sum $V = \mathbb{R}^{\coprod G/H_i}$ for a collection of finite index subgroups $H_i < G$ such that there is a bijection between the orbits $X_i$ of $G$ in $X$ and the quotients $G/H_i$. □

Note that in general a permutation representation is given by the action of a group $G$ on a set $X$ and then extended on $\mathbb{R}^X$ such that, for each $g \in G$ and $x \in X$, an element $e_x$ of the canonical basis of $\mathbb{R}^X$ transforms as $g e_x = e_{gx}$. But the action of $G$ on $X$, i.e., the computation of $gx$, is not explicit and easy to convert in computational terms, while the action of $G$ on one of its quotients $G/H$ provides an algebraic, hence computable, alternative. Furthermore, this notion provides the complete list of possible sets admitting $G$-actions up to isomorphism, which is in bijection with the set of all the quotients of $G$.

Let us now discuss how representation spaces of permutations equivariant networks on sets (Zaheer et al., 2017; Qi et al., 2017) reduce to a simple instance of the representation space shown in Corollary 1. Indeed, the symmetric group of $n$ elements, $S_n$, is the set of permutations of $[n]$, and then $S_n$-equivariant networks are able to process sets of elements independently of their order. A complete treatment of the representations of $S_n$ can be found in Sagan (2001). Now we want to construct group quotients able to define representations for $S_n$-equivariant networks used in practice. Consider $\lambda = (\lambda_1, \dots, \lambda_l)$ to be a partition of $n$, i.e., a decreasingly ordered tuple of positive integers whose sum is $n$. Define $S_\lambda = S_{\lambda_1} \times \cdots \times S_{\lambda_l}$ as the subgroup of $S_n$ where the $i$th factor permutes the elements form $\sum_{j=1}^{i-1} \lambda_j$ to $\sum_{j=1}^{i} \lambda_j$. Now set $\lambda$ to be the partition $(n-1, 1)$, elements of $S_n/S_{(n-1,1)}$ will be represented by the identity element and all the permutations of $[n]$ that send 1 into an element $i$ of $\{2, \dots, n\}$ which we indicate with $[(1i)]$ (See Definition 8 for more information about quotients of groups). The action of $\sigma \in S_n$ on $[(1i)] \in S_n/S_{(n-1,1)}$ will be $[(1\sigma(i))]$ where we can identify $[(11)]$ with the identity element. The bijection $[(1i)] \mapsto i$ from $S_n/S_{(n-1,1)}$ to $[n]$ is compatible with the action of $S_n$ on those sets, hence we have an equivariant isomorphism between $\mathbb{R}^{S_n/S_{(n-1,1)}}$ and $\mathbb{R}^n$ which is the standard representation space for permutations equivariant networks on sets (Zaheer et al., 2017; Qi et al., 2017). Due to Corollary 1, $\mathbb{R}^{S_n/S_{(n-1,1)}}$ is one of the admissible representation spaces for $S_n$ on the family of non-odd non-affine continuous activations. In a similar way, in Section 6.2, we will see how to obtain representation spaces of IGNs and we will highlight that many other admissible representation space could be possible. It is relevant how it is possible to obtain efficient models through this procedure and which however offers the possibility of building models starting from the simple knowledge of the group of symmetries of the input.

## 5.2 DISENTANGLED REPRESENTATIONS

Disentangled representations (Cohen & Welling, 2016b) can be described as follows. An irreducible representation of a $G$-representation $V_i$ is a minimal non-trivial $G$-invariant subspace, and each representation $V_i$ can be decomposed into a direct sum of irreducible spaces (Definition 15 and Theorem 5). Composing irreducible representations with each other allows us to construct arbitrary representation spaces and control the number of parameters of each layer at will thanks to Schur's Lemma (Lemma 2). The easiest way to compose these representations with each other is by doing a direct sum. If then, to define activations, we choose a basis whose vectors are contained in a single irreducible component we say that such a space is disentangled (Cohen & Welling, 2016b). As a consequence of Theorem 1, we obtain the following characterization of disentangle representations.

**Corollary 2.** *For finite groups and activation functions $f : V \to V$ induced by non-odd non-affine continuous functions, the representation is disentangled if and only if the representation $V$ is a sum of trivial representations.*

*Proof.* The direct sum of trivial representations is clearly disentangled and admissible for Theorem 1. To prove the inverse implication, thanks to Corollary 1, we can consider $V$ to be a permutation representation. By disentanglement, we can suppose $V$ to be irreducible and with basis

$\mathcal{B} = \{v_1, \ldots, v_m\}$ defining $f$. Given $v \in V$, the subspace $\langle \sum_{g \in G} gv \rangle$ is trivial, non-zero because $V$ is a permutations representation, and $G$-invariant. Hence, $V = \langle \sum_{g \in G} gv \rangle$ is trivial. $\qquad \square$

Even if composability and control of the number of parameters are particularly good properties of disentangled network, being able to admit only trivial irreducible representations is not achievable in most cases of practical use. Let us consider again the example of representation spaces of permutation equivariant networks on sets (Zaheer et al., 2017; Qi et al., 2017). The irreducible decomposition of the standard action of $S_n$ on $\mathbb{R}^n$ is the direct sum of a trivial component and a $(n-1)$-dimensional one. Hence, a linear layer between this input space and a disentangled admissible representation space will send the input contained in the $(n-1)$-dimensional component to $0$ due to Schur's Lemma (Lemma 2). For large $n$ as customary, this eliminates the entire information inside the input in the forward pass of the first layer of the network.

## 6 PRACTICAL USE CASES

In this section, we aim at understanding how Theorem 2 affects and limits the design choices of networks in a relevant practical scenario such as geometric IGNs (Joshi et al., 2023), a broad family of models particularly proficient in processing geometric data such as point-clouds or meshes. We achieve this objective in Section 6.3, but beforehand, we need to decompose our task into two more manageable subproblems. Specifically, in Section 6.1, we investigate rotation-equivariant networks, a case of networks of that will be necessary to understand the following analysis. In Section 6.2, we delve into higher-order IGN, widely employed in practical applications. Finally, in Section 6.3, we merge the preceding results to analyze geometric IGN comprehensively.

### 6.1 ROTATION EQUIVARIANCE AND EQUIVARIANCE FOR COMPACT GROUPS

Rotation-equivariant neural networks (Finkelshtein et al., 2022) have the ability to process geometrical data tracking orientation and pose. We now discuss how our results affects the design of this kind of networks. The group of rotations around the origin of $\mathbb{R}^3$ is denoted as $\mathrm{SO}(3)$, it can be described as the group of real orthogonal $3 \times 3$ matrices with positive determinant. More generally, let $G = \mathrm{SO}(n)$ be the group of real orthogonal $n \times n$ matrices with positive determinant. It acts on $\mathbb{R}^n$ by left multiplication. Note that $\mathrm{SO}(n)$ is a connected and compact topological group and the presented representation is irreducible and non-trivial (Fulton & Harris, 2004). The following Corollary 3 implies that rotation-equivariant networks are invariant.

**Corollary 3.** *Let $G$ be compact group and let $G_0$ be the connected component containing the identity element. An admissible $G$-representation for non-affine activation functions is $G_0$-invariant. In particular, if $G$ is connected, an admissible $G$-representation is trivial.*

*Proof.* First, suppose $G$ to be connected. The image of a continuous representation of a compact and connected topological group is a compact and connected matrix group due to Theorem 4. The only possible representations described by Theorem 2 are permutation representations and signed permutation representations whose images are finite groups whose only compact and connected subgroup is the one containing only the identity. Hence, the original representation is trivial. If $G$ is not connected, a $G$-representations induced a $G_0$-representations which will be $G_0$-invariant. $\qquad \square$

This means that in general it will be possible to create neural networks capable of performing invariant tasks such as classification but not more general equivariant tasks such as segmentation or detection (Qi et al., 2017). Further, as $\mathrm{SO}(3)$ is a subgroup of Euclidean transformations of $\mathbb{R}^3$, this phenomenon afflicts networks equivariant with respect to general Euclidean transformations, i.e., they will be invariant to the rotational part of Euclidean transformations although possibly sensitive to reflections and translations.

### 6.2 ON INVARIANT GRAPH NETWORKS

Let us now go back at the example proposed in Section 5.1 and generalize it to high-order structures to obtain IGNs. Introduced by Maron et al. (2018), they are a class of neural networks equivariant with respect to the symmetric group and their expressiveness is intimately related to graph neural

networks (Maron et al., 2019a; Geerts, 2020). They are permutation equivariant models taking as input a relational structure such as a set, a graph, or an higher-order structure such as simplicial complexes in form of a tensor of the corresponding order. For example, a directed graph with $n$ nodes can be seen as a tensor in $\mathbb{R}^n \otimes \mathbb{R}^n$. Elements of this space can be represented as linear combinations of $e_i \otimes e_j$ for $i, j \in [n]$ and each $\sigma \in S_n$ acts on them permuting their indices simultaneously, $\sigma(e_i \otimes e_j) = e_{\sigma(i)} \otimes e_{\sigma(j)}$. More in general, a $k$-ary relational structure can be represented as a $k$-order tensor in $(\mathbb{R}^n)^{\otimes k}$ with $S_n$ simultaneously acting on each $\mathbb{R}^n$ component. Regardless of the order of the input tensor, intermediate representation spaces may be the sum of tensors of arbitrary order, hence a linear layer of an IGN will be the direct sum of linear equivariant maps between spaces $(\mathbb{R}^n)^{\otimes k}$ and $(\mathbb{R}^n)^{\otimes h}$ for arbitrary $k$ and $h$ and they admit point-wise activations, hence, by Corollary 1, they should be able to be represented as $\mathbb{R}^{\prod_i S_n/H_i}$ for some subgroups $H_i < S_n$. In Section 5.1 we have seen how $\mathbb{R}^n \cong \mathbb{R}^{S_n/S_{(n-1,1)}}$. Following Benkart et al. (2016), we get $(\mathbb{R}^n)^{\otimes k} = \oplus_{t=0}^k a_t \mathbb{R}^{S_n/S_{(n-t,1^t)}}$ where $a_t$'s are positive integers.

This shows that representation spaces of IGNs are direct sum of $\mathbb{R}^{S_n/S_{(n-t,1^t)}}$ for some $t$. But such $(n-t, 1^t)$ are only but a fraction of the partitions of $n$, therefore $S_{(n-t,1^t)}$'s are only but a fraction of the subgroups $S_\lambda$ which are only few of the subgroups of $S_n$ as they are not transitive on $[n]$ for $\lambda \neq (n)$, unlike other subgroups such as the alternating group, $A_n$. Indeed, the module $\mathbb{R}^{S_n/A_n}$ will be a two-dimensional representation compatible with point-wise activations. Maron et al. (2019a) and Geerts & Reutter (2022) prove lower and upper bounds on the expressiveness of $k$-order IGNs. In the proofs of these bounds, they implicitly employ the decomposition $(\mathbb{R}^n)^{\otimes k} = \oplus_{t=0}^k a_t \mathbb{R}^{S_n/S_{(n-t,1^t)}}$ as this equality is strongly related to the decomposition in tensors of different partition type. This makes it natural to ask what the expressiveness of models employing other types of admissible spaces would be.

## 6.3 Geometric Graphs and Product Groups

Geometric graphs (Joshi et al., 2023; Finkelshtein et al., 2022; Bekkers et al., 2023; Gasteiger et al., 2022) are utilized as a data structure for modeling systems in computational biology, computational chemistry, and computer graphics. Those are graphs, or higher-order structures, embedded in the Euclidean space and as such may transform according to the symmetries of the ambient space, i.e. isometries of $\mathbb{R}^3$, $E(3)$. Hence, it becomes relevant to develop neural networks simultaneously equivariant to such ambient symmetries and node permutations, and in algebraic terms this means that we need $E(3) \times S_n$-equivariant networks able to process elements in $\mathbb{R}^3 \otimes (\mathbb{R}^n)^{\otimes k}$, where the first tensor factor represent the geometrical features and the second the relational structure. We synthesize the results concerning this case in the following corollary of Theorem 2. See Appendix A.6 for the proof.

**Corollary 4.** *Every* $\mathrm{SO}(3) \times S_n$*-equivariant layer defined on* $\mathbb{R}^3 \otimes (\mathbb{R}^n)^{\otimes k}$ *coupled with non-affine activations is null. Hence,* $E(3) \times S_n$*-equivariant networks are rotation invariant.*

## 7 Conclusions and Future Directions

In conclusion, we have provided a complete characterization of networks with equivariant layers featuring point-wise equivariant activations up to isomorphism of the linear part. Our analysis investigates relevant examples, including IGNs and rotation-equivariant CNNs. Future work will involve investigating equivalent results for complex-valued networks, as this extension is made possible by the novel algebraic approach that we have introduced for the proof of Theorem 1. Since complex numbers admit non-dense non-discrete multiplicative subgroups, this property should alleviate the limitations of real-valued networks equivariant with respect to non-discrete groups. Future research directions beyond point-wise activations introduce us to a wide domain with no explicit guidelines for navigation. Various endeavors have been documented in the literature, some of which are norm activations (Worrall et al., 2017), squashing activations (Sabour et al., 2017), tensor product activations (Kondor, 2018), and gated activations (Weiler et al., 2018). An instance of activation functions amenable to theoretical analysis can be found in equivariant polynomial functions. Structures–such as symmetric polynomials–have been studied in mathematics for a long time and are beginning to provide insights for the construction of efficient equivariant models (Puny et al., 2023).

## 8  ACKNOWLEDGEMENT

Bruno Lepri acknowledges the support of the PNRR project FAIR - Future AI Research (PE00000013), under the NRRP MUR program funded by the NextGenerationEU and the support of the European Union's Horizon Europe research and innovation program under grant agreement No. 101120237 (ELIAS).

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

# A APPENDIX

## A.1 TOPOLOGICAL SPACES

**Definition 2.** *A* topological space[1] *is a pair* $(X, \tau)$*, where* $X$ *is a set and* $\tau$ *is a collection of subsets of* $X$ *such that:*

1. *$X$ belongs to $\tau$.*

2. *The intersection of any finite number of sets in $\tau$ is also in $\tau$.*

3. *The union of any collection of sets in $\tau$ is also in $\tau$.*

*The sets in $\tau$ are called* open sets*, and the collection $\tau$ is called the* topology *on $X$.*

**Example 1** (Discrete Topology)**.** *The* discrete topology *on a set $X$ is the topology in which every subset of $X$ is an open set, i.e.,* $\tau = \mathcal{P}(X)$*, where* $\mathcal{P}(X)$ *is the power set of $X$.*

*The set of integers $\mathbb{Z}$ with the discrete topology is a topological space. In this topology, every singleton set $\{n\}$ is open for all $n \in \mathbb{Z}$.*

**Definition 3** (Continuous Maps)**.** *Let* $(X, \tau_X)$ *and* $(Y, \tau_Y)$ *be topological spaces. A function* $f : X \to Y$ *is said to be* continuous *if for every open set $V$ in $Y$, the inverse image $f^{-1}(V)$ is an open set in $X$.*

**Definition 4** (Compact Spaces)**.** *An* open cover *of a topological space $X$ is a set $\mathcal{U} \subseteq \tau$ such that* $X = \bigcup_{U \in \mathcal{U}} U$. *A* refinement *of a cover $\mathcal{U}$ of $X$ is a subset of $\mathcal{U}$ which is still a cover. A topological space $X$ is said to be* compact *if every open cover of $X$ has a finite refinement.*

**Theorem 3.** *A subspace $X \subseteq \mathbb{R}^n$ is compact if and only if is closed and limited.*

**Definition 5** (Connected Spaces)**.** *A topological space $X$ is said to be* connected *if it cannot be expressed as the union of two disjoint nonempty open sets. A subset of $X$ who is maximal under containment order and connected is called a* connected component *of $X$.*

**Theorem 4.** *If $X$ and $Y$ are two topological and $f : X \to Y$ is a continuous map then the two following statements hold*

- *if $X$ is compact then $f(X)$ is compact*

- *if $X$ is connected then $f(X)$ is connected*

## A.2 GROUP THEORY

**Definition 6.** *A **group** is a pair $(G, \cdot)$ where $G$ is a set and $\cdot : G \times G \to G$ is a function satisfying the following axioms.*

- ***Associativity:** for each $g, h, k \in G$ we have$(g \cdot h) \cdot k = g \cdot (h \cdot k)$.*

- ***Identity:** there exists an element $e \in G$ such that $g \cdot e = e \cdot g = g$ for each $g \in G$.*

- ***Inverse Element:** for each element $g \in G$, there exists an element $g^{-1} \in G$ such that $g \cdot g^{-1} = g^{-1} \cdot g = e$.*

*A group is **finite** if it contains a finite number of elements. A group is **abelian** or **commutative** if $gh = hg$ for each $g, h \in G$.*

---

[1]Aggiungere ref, va bene Hatcher?

**Example 2.** *Here we present some fundamental examples of groups.*

- *The set of integers with addition.*

- *The set $\mathbb{S}^1 = \{\rho_\alpha\}$ of rotations of angle $\alpha$ centered in the origin of the 2D Cartesian plane with composition.*

- *The set $\mathrm{GL}(V)$ of bijective linear maps of a vector space $V$ into itself with composition. Then, the set $\mathrm{GL}_n(\mathbb{R})$ of $n \times n$ real invertible matrices form a group where the operation is row-column multiplication.*

- *Let $\mathrm{SO}_n(\mathbb{R})$, or simply $\mathrm{SO}(n)$, be the group of orthogonal matrices with positive determinant.*

- *Fix $[n] = \{1, \ldots, n\}$. The set*

$$S_n = \{f : [n] \to [n] \mid f \text{ is bijective}\}$$

*with the composition operation form the **symmetric group** or the **permutation group**.*

- *Given two groups $G$ and $H$, the direct product $G \times H$ of them is still a group. The set of the elements is the Cartesian product of $G$ and $H$ while the sum is defined as*

$$(g_1, h_1) \circ_{G \times H} (g_2, h_2) = (g_1 \circ_G g_2, h_1 \circ_H h_2).$$

Now, we introduce notion of group homomorphism, a transformation between groups which preserves the operation.

**Definition 7.** *A **group homomorphism** is a map*

$$\phi : G \to H$$

*between $G$ and $H$ groups such that, for each $g, h \in G$*

$$\phi(g \cdot h) = \phi(g) \cdot \phi(h).$$

**Example 3.** *The map $\Phi : \mathbb{S}^1 \to \mathrm{GL}_2(\mathbb{R})$ defined by*

$$\rho_\alpha \mapsto \begin{bmatrix} \cos(\alpha) & \sin(\alpha) \\ -\sin(\alpha) & \cos(\alpha) \end{bmatrix} \tag{2}$$

*is an homomorphism between the group of rotations of angle $\alpha$ and the $2 \times 2$ invertible matrices.*

**Definition 8** (Quotients). *Let $G$ be a group and $H$ be a subgroup of $G$. The* quotient *of $G$ by $H$ is the set $G/H = \{gH : g \in G\}$, where $gH = \{gh : h \in H\}$ are the* left cosets *of $H$.*

**Example 4.**      *1. Consider $G = \mathbb{Z}$ and the subgroup $H = n\mathbb{Z}$ of integers multiples of $n$. The quotient $G/H$ is a group and is isomorphic to the cyclic group of $n$ elements, $\mathbb{Z}_n$, i.e., integers modulo $n$.*

     *2. Consider $G = S_3$, the symmetric group on three elements, and the subgroup $H = \{(1), (12)\}$. The quotient $G/H$ is a group and is isomorphic to $S_2$, symmetric group on two elements.*

     *3. Consider $G = S_n$, the symmetric group on $n$ elements, and the subgroup $H = A_n$, the alternating group on $n$ elements. The quotient $G/H$ is a group and is isomorphic to $\mathbb{Z}_2$.*

## A.3    TOPOLOGICAL GROUPS

**Definition 9.** *A* topological group *is a group $G$ equipped with a topology such that multiplication and inversion are continuous maps.*

**Example 5.**      - *The real numbers $\mathbb{R}$ with the standard topology form a topological group under addition.*

- *The multiplicative group of non-zero real numbers $\mathbb{R}^*$ with the standard topology is a topological group.*

- *Each finite group $G$ with the discrete topology is a topological group. Note that the connected components of this group are all the singleton elements $\{g\}$ for each $g \in G$. Note that this group is compact as $\mathcal{P}(G)$ is finite, hence each open cover is finite.*

- *The additive group of integers $\mathbb{Z}$ with the discrete topology is a topological group. Note that the connected components of this group are all the singleton elements $\{n\}$ for each $n \in \mathbb{Z}$. All this singletons form an infinite open cover of $\mathbb{Z}$ which cannot be refined.*

## A.4 Representation Theory

**Definition 10.** *A **representation** of a group $G$ in a vector space $V$ is a group homomorphism*
$$\rho : G \to \mathrm{GL}(V).$$

**Definition 11.** *Consider $G = S_n$.*

- *$\dim V = 1$ and $\rho(g) = id$ for each $g \in S_n$. This is called the **trivial** representation of $S_n$.*

- *$\dim V = 1$ and $\rho(g) = sgn(g)id$. This is called the **sign** representation of $S_n$.*

**Definition 12.** *A representation $\rho : G \to \mathrm{GL}_n(\mathbb{R})$ is*

- ***Non-negative** if all the elements of each $\rho(g)$ are non-negative.*

- ***Monomial** if each row and column of each $\rho(g)$ has exactly one non-zero element.*

- *A **permutation** representation if it is monomial and each non-zero element is $1$.*

- *A **signed permutation** representation if it is monomial and each non-zero element is $\pm 1$.*

**Definition 13** (Representations of Topological Groups)**.** *Let $G$ be a topological group and $V$ be real vector space. A continuous representation of $G$ on $V$ is a continuous group homomorphism $\rho : G \to GL(V)$.*

**Example 6.** *Consider the general linear group $\mathrm{GL}(V)$ of invertible $n \times n$ matrices, with the topology induced by the Euclidean norm on matrices. Let $V = \mathbb{R}^n$ be the standard Euclidean space. The map $\rho : G \to \mathrm{GL}(V)$ defined by $\rho(A)(v) = Av$ is a continuous representation. Similarly, restricting this representation to $\mathrm{SO}(n)$ we get a representation of $\mathrm{SO}(n)$.*

**Definition 14.** *A linear map $\Phi : V \to W$ is $G$-**equivariant** with respect to the representations $\rho_1 : G \to \mathrm{GL}(V)$ and $\rho_2 : G \to \mathrm{GL}(W)$ if*
$$\rho_2 \circ \Phi = \Phi \circ \rho_1.$$
*We will denote the space of all $G$-equivariant maps between $\rho_1$ and $\rho_2$ by $Hom_G(\rho_1, \rho_2)$ or $Hom_G(V, W)$ when $\rho_1$ and $\rho_2$ will be clear from the context.*

**Definition 15.** *A representation $\rho : G \to \mathrm{GL}(V)$ is **irreducible** if there exists no non-trivial subspace $W$ of $V$ such that $\rho(g)(W) \subseteq W$ for each $g \in G$.*

An important result in representation theory of finite groups states that there is always a decomposition into irreducible representations.

**Theorem 5.** *For each representation $\rho : G \to \mathrm{GL}(V)$, there exists a decomposition*
$$V = V_1 \oplus \cdots \oplus V_m$$
*where each $V_i$ is irreducible for $\rho$. This decomposition is unique up to isomorphism and permutation of the factors.*

Another tool coming from Representation Theory is Schur's Lemma which will be fundamental to understand our results.

**Lemma 2.** *Let $V$ and $W$ be non-isomorphic irreducible representations, there is only one $G$-equivariant linear map between them and it is the trivial one.*

**Definition 16.** *The fixed set for a representation $\rho : G \to \mathrm{GL}(V)$ is $V^G = \{v : gv = v\}$. Note that $V^G$ is a representation for $G$ and the action is trivial.*

**Theorem 6.** *The following properties are true for representations of a finite group $G$.*

- *$\dim V^G$ is the multiplicity of the trivial representation in $V$,*

- *$\dim \mathrm{Hom}_G(V, W) = \dim(V \otimes W)^G$.*

## A.5 Affine maps

**Definition 17.** *Let $V$ and $W$ be two $\mathbb{K}$-vector spaces and define the translation of a vector $w$ in $W$ as a non-linear bijective map $\tau_w : v \mapsto v + w$. Define the space of **affine maps** from $V$ to $W$ as*

$$\text{Aff}(V, W) = \{\tau_w \circ f | w \in W \text{ and } f \in \text{Hom}(V, W)\}.$$

Note that is a more general definition with respect to the standard one, where $f$ is an isomorphism of a vector space $V$.

**Theorem 7.** *The decomposition of an affine map $\phi \in \text{Aff}(V, W)$ in translational part $\tau_w$ and $f$ linear part is unique.*

*Proof.*

$$\tau_{w_1} \circ f_1 = \phi = \tau_{w_2} \circ f_2,$$

evaluating in 0 leads to

$$w_1 = \phi(0) = w_2.$$

Write $w = w_1 = w_2$, and note that

$$\tau_w \circ f_1 = \tau_w \circ f_2,$$

by the bijectivity of translations,

$$f_1 = f_2.$$

$\square$

Let $V$ and $W$ be $G$-representation. An affine map $\phi \in \text{Aff}(V, W)$ is $G$-equivariant if $\phi \circ g = g \circ \phi$ for each $g \in G$, write the set of $G$-equivariant affine maps from $V$ to $W$ as $\text{Aff}_G(V, W)$.

**Theorem 8.** $\phi = \tau_w \circ f \in \text{Aff}_G(V, W)$ *if and if $f \in \text{Hom}_G(V, W)$ and $v$ is invariant.*

*Proof.* Note that for each $g \in G$,

$$g \circ \tau_w \circ f = \tau_{gw} \circ (g \circ f).$$

Observe that

$$\phi \circ g = g \circ \phi,$$

if and only if

$$\tau_w \circ f \circ g = g \circ \tau_w \circ f = \tau_{gw} \circ (g \circ f)$$

if and only if, by the previous proposition,

$$w = gw$$

and

$$f \circ g = g \circ f$$

for each $g \in G$. $\square$

## A.6 Representations of Group Products on Tensor Products

**Remark 1.** *Let $V \otimes W$ be a finite-dimensional $G \times H$-representations and $V_i$'s a complete list of irreducible $G$-representations and $W_j$'s a complete list of irreducible $H$-representations, then $V_i \otimes W_j$ 's is a complete list of irreducible $(G \times H)$-representations (See Sagan (2001) for proofs in case $G$ and $H$ are finite). If $m_i$ is the multiplicity of $V_i$ in $V$ and $n_j$ is the multiplicity of $W_j$ in $W$ then the multiplicity of $V_i \otimes W_j$ in $V \otimes W$ is $m_i n_j$. This can be easily seen by writing the irreducible decompositions of $V$ and $W$ and use the distributive property of tensor products and direct sums. Note that the same is true if $G$ and $H$ are compact groups and the representations are continuous. If $S$ is an $G$-isotypic component of $V \times W$ of type $\rho$ then it is $H$-invariant and each $h \in H$ acts as $G$-equivariant endomorphism of $S$. By Lemma 2, $S = \rho \otimes \sigma$, which is $\bigoplus_i \rho \otimes \sigma_i$, where $\sigma_i$ are irreducible $H$-representations. Iterating the decomposition if necessary and by the finite dimension of $V$ and $W$, we conclude.*

Thanks to these observations we can prove Corollary 4

*Proof.* Let us study $SO(3) \times S_n$-equivariant linear layers. Irreducible representations of $SO(3) \times S_n$ are the tensor products of irreducible representations of $SO(3)$ and $S_n$ as shown above. The natural action of $SO(3)$ on $\mathbb{R}^3$ is an irreducible representation and admissible irreducible representations are invariant, i.e., the trivial action of $SO(3)$ on $\mathbb{R}$. Hence, the fist layer of an $SO(3) \times S_n$-equivariant network processing a geometric graph will be a linear layer from a direct sum of $\mathbb{R}^3 \otimes S^\lambda$ to a direct sum of $\mathbb{R} \otimes S^\mu$, where $S^\lambda$ and $S^\mu$ indicate irreducible representations for $S_n$. But, due to Schur's Lemma, the only possible linear layer would be the null one, i.e., a layer without trainable parameters. □

### A.7 PROOF OF THE STABILIZATION LEMMA

For convenience we restate Lemma 1.

**Lemma 3.** *The group of matrices $\mathcal{M}' = \mathcal{M}(\mathcal{F}(\mathcal{M}))$ is the largest group in $\mathrm{GL}_n(\mathbb{R})$ for which $\mathcal{F}(\mathcal{M}') = \mathcal{F}(\mathcal{M})$, and $\mathcal{F}' = \mathcal{F}(\mathcal{M}(\mathcal{F}))$ is the largest family of functions in $\mathcal{C}(\mathbb{R})$ for which $\mathcal{M}(\mathcal{F}') = \mathcal{M}(\mathcal{F})$.*

In what follows we state and prove some results necessary for the proof of Lemma 1. The proof of the next lemma is trivial.

**Lemma 4.** *The two following statements are true.*

1. *For each group of matrices $\mathcal{M}_1$, $\mathcal{M}_1 \subseteq \mathcal{M}(\mathcal{F}(\mathcal{M}_1))$ and for each family of functions $\mathcal{F}_1$, $\mathcal{F}_1 \subseteq \mathcal{F}(\mathcal{M}(\mathcal{F}_1))$,*

2. *For each inclusion $\mathcal{M}_1 \subseteq \mathcal{M}_2$ of groups of matrices, $\mathcal{F}(\mathcal{M}_2) \subseteq \mathcal{F}(\mathcal{M}_1)$, similarly, for families of activations $\mathcal{F}_1 \subseteq \mathcal{F}_2$, $\mathcal{M}(\mathcal{F}_2) \subseteq \mathcal{M}(\mathcal{F}_1)$.*

**Lemma 5.** *For each family of functions $\mathcal{F}_1$, $\mathcal{M}(\mathcal{F}_1) = \mathcal{M}(\mathcal{F}(\mathcal{M}(\mathcal{F}_1)))$. For each group of matrices $\mathcal{M}_1$, $\mathcal{F}(\mathcal{M}_1) = \mathcal{F}(\mathcal{M}(\mathcal{F}(\mathcal{M}_1)))$.*

*Proof.* We prove the equality $\mathcal{M}_1$, $\mathcal{M}(\mathcal{F}_1) = \mathcal{M}(\mathcal{F}(\mathcal{M}(\mathcal{F}_1)))$ first by proving $\mathcal{M}_1$, $\mathcal{M}(\mathcal{F}_1) \subseteq \mathcal{M}(\mathcal{F}(\mathcal{M}(\mathcal{F}_1)))$, then we prove the opposite inclusion.

Substituting $\mathcal{M}_1$ with $\mathcal{M}(\mathcal{F}_1)$ in the first point of Lemma 4, we get $\mathcal{M}(\mathcal{F}_1) \subseteq \mathcal{M}(\mathcal{F}(\mathcal{M}(\mathcal{F}_1)))$.

To prove the opposite inclusion, substitute $\mathcal{F}_2 = \mathcal{F}(\mathcal{M}(\mathcal{F}_1))$ in the second point of Lemma 4, we obtain $\mathcal{M}(\mathcal{F}(\mathcal{M}(\mathcal{F}_1))) \subseteq \mathcal{M}(\mathcal{F}_1)$. Implying $\mathcal{M}(\mathcal{F}_1) = \mathcal{M}(\mathcal{F}(\mathcal{M}(\mathcal{F}_1)))$. In a similar way, we can prove the other equality. □

We are now ready to prove Lemma 1.

*Proof.* We only prove the first equality as the second have an analogous proof. By Lemma 5, we know that $\mathcal{F}(\mathcal{M}') = \mathcal{F}(\mathcal{M})$. Then, for each $\mathcal{S} \subseteq \mathrm{GL}_n(\mathbb{R})$ such that $\mathcal{F}(\mathcal{S}) = \mathcal{F}(\mathcal{M})$, we know that $\mathcal{S} \subseteq \mathcal{M}(\mathcal{F}(\mathcal{S}))$ by Lemma 4. Moreover, $\mathcal{S} \subseteq \mathcal{M}(\mathcal{F}(\mathcal{S})) = \mathcal{M}(\mathcal{F}(\mathcal{M})) = \mathcal{M}'$. Hence each $\mathcal{S} \subseteq \mathrm{GL}_n(\mathbb{R})$ such that $\mathcal{F}(\mathcal{S}) = \mathcal{F}(\mathcal{M})$ we have $S \subseteq \mathcal{M}'$, therefore $\mathcal{M}' = \mathcal{M}(\mathcal{F}(\mathcal{M}))$ is the largest group in $\mathrm{GL}_n(\mathbb{R})$ such that $\mathcal{F}(\mathcal{M}') = \mathcal{F}(\mathcal{M})$. □

### A.8 PROOF OF THE MAIN THEOREM

We can now state our main result, which shows that, under mild assumptions, there are only a very limited number of maximal groups (or equivalently, of maximal function classes). Moreover, for each of these we provide the exact correspondence between $\mathcal{M}$ and $\mathcal{F}$, thus providing an exhaustive classification of all possible admissible pairs of $\mathcal{M}$ and $\mathcal{F}$.

A class of functions fundamental for the understating of what follows will be $T$-multiplicative functions which we define as follows.

**Definition 18.** *Let $T$ be a multiplicative subgroup of $\mathbb{R}$. We say that a continuous function $f$ is $T$-multiplicative if $f(tx) = tf(x)$ for each $t \in T$ and $x \in \mathbb{R}$. We write $\mathcal{F}_T$ to indicate the set of all continuous $T$-multiplicative functions.*

Note that if $T = \langle b^n \rangle_{n \in \mathbb{Z}}$, the notion of $T$-multiplicativity reduces to the notion of $b$-multiplicativity provided in Section 4. Similarly, for $T = \langle \pm b^n \rangle_{n \in \mathbb{Z}}$ and $\pm b$-multiplicativity.

We are now ready to state Theorem 9, one of the two key results fundamental to prove Theorem 1. In particular, Theorem 9 characterizes admissible pairs indexing them as the multiplicative subgroups of $\mathbb{R}$ but does not provide a constructive description of their families of activation functions, this description is given by Lemma 10. In what follows we will write $\mathcal{R}_n$ for the set of all unit row invertible matrices.

**Theorem 9.** *Maximal admissible pairs for $T = \mathcal{T}(\mathcal{M})$ are*

- $(\mathcal{M}, \text{Aff}(\mathbb{R}, \mathbb{R}))$ *for each group of non-monomial matrices $\mathcal{M}$ in $\mathcal{R}_n(\mathbb{R})$ and $(\text{GL}_n(\mathbb{R}), \text{Hom}(\mathbb{R}, \mathbb{R}))$ if $T$ is dense,*

- $(\mathcal{M}_n(T), \mathcal{F}_T)$ *otherwise.*

Let us denote $\mathbb{R}^*$ as the set of non-zero reals and note that it is a multiplicative group. In the following, we will use the following characterization of the multiplicative subgroups of $\mathbb{R}^*$ that we need to prove Lemma 10.

**Lemma 6.** *Multiplicative subgroups of $\mathbb{R}_{>0}$ are the trivial group $\langle 1 \rangle$, discrete unbounded groups $\langle b^n \rangle_{n \in \mathbb{Z}}$, and dense subgroups of $\mathbb{R}_{>0}$. Multiplicative subgroups of $\mathbb{R}^*$ are the trivial one, $\langle 1 \rangle$, the other finite one, $\langle \pm 1 \rangle$, discrete positive ones, $\langle b^n \rangle_{n \in \mathbb{Z}}$, the other discrete ones, $\langle \pm b^n \rangle_{n \in \mathbb{Z}}$, the ones dense on $\mathbb{R}_{>0}$, and dense ones.*

*Proof.* Multiplicative subgroups of $\mathbb{R}_{>0}$ and additive subgroups of $\mathbb{R}$ are linked by the exponential map which is an isomorphism. Additive subgroups of $\mathbb{R}$ are divided into three different type: finite ($\langle 0 \rangle$), unbounded and discrete ($\alpha \mathbb{Z}$ for each $\alpha \in \mathbb{R}_{>0}$) and dense. They map through into $\mathbb{R}_{>0}$ as finite ($\langle 1 \rangle$), unbounded and discrete ($\{ b^n = e^{n\alpha} \}_{n \in \mathbb{Z}} \cong \mathbb{Z}$) and dense. We can get the multiplicative subgroups of $\mathbb{R}^*$ noticing that $\mathbb{R}^* \cong \mathbb{Z}_2 \times \mathbb{R}_{>0}$. $\square$

The following technical lemmas will be necessary to prove Lemma 10.

**Lemma 7.** *Let $b$ a real number, $b > 1$, and $T = \langle b^n \rangle_{n \in \mathbb{Z}}$. For each positive real number $x$ there exist a unique decomposition $x = b^n y$ such that $y \in [1, b)$ and $n \in \mathbb{Z}$.*

*Proof.* Choose $n$ such that $x \in [b^n, b^{n+1})$, we have that $x = b^n y$ where we define $y = \frac{x}{b^n} \in [1, b)$. Sets $\{ [b^n, b^{n+1}) \}_{n \in \mathbb{Z}}$ are a collection of disjoint intervals covering $\mathbb{R}$. Hence $x$ belongs only to one of those, say $x \in [b^n, b^{n+1})$, we have the uniqueness of the decomposition $x = b^n y$ such that $y \in [1, b)$. $\square$

Note that Lemma 7 implies that for each $x \in \mathbb{R}_{>0}$ there exist a unique $n \in \mathbb{Z}$ such that $\frac{x}{b^n} \in [1, b)$. This observation grants the following notion to be well-defined. Let $\eta : [1, b] \to \mathbb{R}$ be the function such that $\eta(b) = b\eta(1)$. Define $f_\eta : \mathbb{R}_{>0} \to \mathbb{R}$ as the function $f_\eta(x) = b^n \eta(\frac{x}{b^n})$ where $n$ is the only integer such that $\frac{x}{b^n} \in [1, b)$. Clearly $f_\eta$ is $T$-multiplicative for $T = \langle b^n \rangle_{n \in \mathbb{Z}}$ where $b$ is a real number greater than 1. We now state and prove Lemma 8 which gives a complete description of $T$-multiplicative functions for $T = \langle b^n \rangle_{n \in \mathbb{Z}}$. Furthermore, it provides a constructive procedure offering a computational implementation of such functions.

**Lemma 8.** *Let $b > 1$ be a real number, and let $T = \langle b^n \rangle_{n \in \mathbb{Z}}$. A continuous function $f$ is $T$-multiplicative if and only if there exist two continuous functions $\eta_\pm : [1, b] \to \mathbb{R}$ such that $\eta_\pm(b) = b\eta_\pm(1)$ and*

$$f(x) = \begin{cases} f_{\eta_+}(x) & x > 0 \\ 0 & x = 0 \\ f_{\eta_-}(-x) & x < 0 \end{cases}. \tag{3}$$

*Proof.* ($\Rightarrow$) Define $\eta_\pm = f_{|[\pm 1, \pm b]}$. Multiplicativity and continuity of $f$ implies all the required properties of $\eta_\pm$.
($\Leftarrow$) We only need to prove that $f$ constructed in this way is continuous on $\mathbb{R}$. We will prove the

continuity of $f$ on $\mathbb{R}_{>0}$ and $\mathbb{R}_{<0}$, and in $0$. The two former cases boil down to proving continuity of $f_{\eta_\pm}$ respectively. As those proofs are analogous, we only present the one for $f_{\eta_+}$. Note that for each $x \in \mathbb{R}_{>0} \setminus T$, there exists an interval $x \in I \subseteq \mathbb{R}_{>0} \setminus T$ and an integer $n$ such that $f_{\eta_+}(x) = b^n \eta(\frac{x}{b^n})$ for each $x \in I$. Hence $f_{\eta_+}$ is continuous on $\mathbb{R}_{>0} \setminus T$. Now see that

$$\lim_{x \to b^{n+}} f_{\eta_+}(x) = \lim_{x \to 1^+} b^n \eta(x) = \lim_{x \to b^-} b^{n-1}\eta(x) = \lim_{x \to b^{n-}} f_{\eta_+}(x)$$

because $\eta_+(b) = b\eta_+(1)$. It remains to prove continuity of $f$ in $0$, i.e.,

$$\lim_{x \to 0^-} f_{\eta_+}(x) = \lim_{x \to 0^+} f_{\eta_-}(x).$$

Note that $\eta_+$ is continuous and limited as it is defined on a compact subset of $\mathbb{R}$, hence we can define $m = \min_{x \in [1,b]} \eta_+(x)$ and $M = \max_{x \in [1,b]} \eta_+(x)$, note that $f_{m \cdot 1_{[1,b]}} \leq f_{\eta_+} \leq f_{M \cdot 1_{[1,b]}}$ where $1_{[1,b]}$ is the constant function with value $1$ defined on $[1,b]$. It is easy to see that $\lim_{x \to 0^+} f_{m \cdot 1_{[1,b]}} = \lim_{x \to 0^+} f_{M \cdot 1_{[1,b]}} = 0$. $\square$

We now state Lemma 9 whose proof will essentially contain the proof of Lemma 10.

**Lemma 9.** *Each $T$-multiplicative continuous function is linear if and only if $T$ is dense in $\mathbb{R}$.*

*Proof.* Note that if $f$ is $T$-multiplicative, by definition, $f(t) = tf(1)$ for each $t \in T$. As $T$ is dense in $\mathbb{R}$, $f$ is linear on its entire domain $\mathbb{R}$ by continuous extension. If $T$ is dense in $\mathbb{R}_{>0}$, $f(t) = tf(1)$ and $f(-t) = tf(-1)$ for each $t > 0$, hence $f$ is semilinear. On the other hand, if $T$ is not dense, suppose $T$ is $\langle 1 \rangle$ or $\langle \pm 1 \rangle$, then $\mathcal{F}_T$ are $\mathcal{C}(\mathbb{R})$ and odd functions respectively, which contain non-linear functions. If $T = \langle b^n \rangle_{n \in \mathbb{Z}}$. Let us now proceed with the construction of a function that is continuous, $T$-multiplicative, and non-linear. Let $\eta_\pm : [1,b] \to \mathbb{R}$ be two continuous bump function and let $f$ be a continuous function as defined in Equation 3, this function is multiplicative and non-linear. This concludes the proof as we have listed all the possible cases of $T$ presented in Lemma 6. $\square$

Thanks to Lemmas 6 and 8, the presented proof of Lemma 9 gives a complete characterization of $T$-multiplicative functions with respect to multiplicative subgroups $T$ of $\mathbb{R}$ which we can state as in the following Lemma 10.

**Lemma 10.** *If $T = \langle 1 \rangle$, then $\mathcal{F}_T = \mathcal{C}(\mathbb{R})$. If $T = \langle \pm 1 \rangle$, then $\mathcal{F}_T = \mathcal{O}(\mathbb{R})$, the odd continuous functions. If $T = \langle b^n \rangle_{n \in \mathbb{Z}}$ for $b > 1$, then $\mathcal{F}_T$ are all the continuous functions described in Equation 3 arising from two continuous $\eta_\pm : [1,b] \to \mathbb{R}$ such that $\eta_\pm(b) = b\eta_\pm(1)$. If $T = \langle \pm b^n \rangle_{n \in \mathbb{Z}}$ for $b > 1$, then $\mathcal{F}_T$ are all the continuous functions defined as in the previous case but $\eta_+ = \eta_-$. If $T$ is dense on $\mathbb{R}_{>0}$, then $\mathcal{F}_T$ are all the semilinear functions. Finally, if $T$ is dense on $\mathbb{R}$, then $\mathcal{F}_T$ are all the linear functions on $\mathbb{R}$.*

To prove Theorem 9 we need to state and prove the two following lemmas. The main ideas behind their proofs are primarily due to Wood & Shawe-Taylor (1996).

**Lemma 11.** *Define the multiplicative group $\mathcal{T}(\mathcal{M}) = \langle \sum_{j \in S} M_{ij} : S \subseteq [n], M \in \mathcal{M}, \in [n] \rangle \setminus \{0\}$ and $\tilde{f}_0(x) = \tilde{f}(x) - \tilde{f}(0)$. Note that $\tilde{f}$ is $\mathcal{M}$-equivariant if and only if $\tilde{f}_0$ is $\mathcal{M}$-equivariant and $\tilde{f}_0(0)$ is a $\mathcal{M}$-invariant vector. In particular, if $\tilde{f}$ is $\mathcal{M}$-equivariant then $f_0$ is $\mathcal{T}(\mathcal{M})$-multiplicative and $f(0) = 0$ or $\mathcal{M} \subseteq \mathcal{R}_n$.*

*Proof.* Note that $\tilde{f}(Mx) = M\tilde{f}(x)$ for each $x \in \mathbb{R}^n$ if and only if $\tilde{f}_0(Mx) = \tilde{f}(Mx) - \tilde{f}(M0) = M\tilde{f}(x) - M\tilde{f}(0) = M\tilde{f}_0(x)$ for each $x \in \mathbb{R}^n$ if and only if $\tilde{f}_0(Mx) = M\tilde{f}_0(x)$ for each $x \in \mathbb{R}^n$ and $M\tilde{f}_0(0) = \tilde{f}_0(0)$. In particular, $M\tilde{f}(0) = \tilde{f}(0)$ if and only if $M1f(0) = 1f(0)$, where $1$ is the vector with all ones, if and only if $f(0) = 0$ or $M \in \mathcal{R}_n$. For each $S \subseteq [n]$ and each $x \in \mathbb{R}$ we have that $f_0(\sum_{j \in S} M_{rj}x) = \sum_{j \in S} M_{rj}f_0(x)$ for each $r \in [n]$, hence $f_0$ is $\mathcal{T}(\mathcal{M})$-multiplicative as $f_0(M_ix) = M_if_0(x)$ for each $x \in \mathbb{R}$ and $i = 1, 2$ then $f_0(M_1M_2x) = M_1M_2f_0(x)$. $\square$

**Lemma 12.** *Let $T = \mathcal{T}(\mathcal{M})$ be a non-dense subset of $\mathbb{R}^*$. Then $\mathcal{F}(\mathcal{M})$ contains non-affine functions if and only if matrices in $\mathcal{M}$ are $T$-monomial.*

*Proof.* ($\Rightarrow$) For each $f \in \mathcal{F}(\mathcal{M})$, $f_0 = f - f(0)$ is $T$-multiplicative and $\tilde{f}_0$ is $\mathcal{M}$-equivariant by Lemma 11. Suppose $\mathcal{M}$ contains a matrix $M$ which fails to be $T$-monomial, without loss of generality we may assume $M_{11} = t_1$ and $M_{12} = t_2$ not zero. Hence, $f_0$ is additive. Indeed, for each $x_1, x_2 \in \mathbb{R}$, we have that

$$f_0(x_1 + x_2) = \langle \tilde{f}_0(M(t_1^{-1}x_1e_1 + t_2^{-1}x_2e_2)), e_1 \rangle =$$

$$\langle M\tilde{f}_0(t_1^{-1}x_1e_1 + t_2^{-1}x_2e_2), e_1 \rangle = t_1 f_0(t_1^{-1}x_1) + t_2 f_0(t_2^{-1}x_2) = f_0(x_1) + f_0(x_2)$$

Therefore $f_0$ is linear, being both additive and continuous, this implies $f$ to be affine which contradicts the hypothesis.

($\Leftarrow$) If $\mathcal{M}$ contains only $T$-monomial matrices, it is easy to check that each $T$-multiplicative function induces an equivariant activation, which are not all affine by Lemma 9. $\qquad\square$

Now we are ready to present the proof of Theorem 9. For convenience in what follows we will write as $\mathcal{P}_n$, the group of $n \times n$ permutation matrices.

*Proof.* We will study which are the groups of matrices $\mathcal{M}$ such that $\mathcal{T}(\mathcal{M}) = T$ as $T$ varies between subgroups of $\mathbb{R}^*$. If $T$ is a dense subgroup of $\mathbb{R}^*$, Lemma 9 and Lemma 11 implies $\mathcal{F}(\mathcal{M}) = \mathrm{Aff}(\mathbb{R}, \mathbb{R})$ if $\mathcal{M} \subseteq \mathcal{R}_n$ or $\mathcal{F}(\mathcal{M}) = \mathrm{Hom}(\mathbb{R}, \mathbb{R})$ otherwise. In those cases, $\mathcal{M}(\mathrm{Aff}(\mathbb{R}, \mathbb{R})) \subseteq \mathcal{R}_n(\mathbb{R})$ and $\mathcal{M}(\mathrm{Hom}(\mathbb{R}, \mathbb{R})) = \mathrm{GL}_n(\mathbb{R})$. For dense $T$, Lemma 1 implies that admissible maximal pairs are $(\mathcal{M} \subseteq \mathcal{R}_n, \mathrm{Aff}(\mathbb{R}, \mathbb{R}))$ and $(\mathrm{GL}_n(\mathbb{R}), \mathrm{Hom}(\mathbb{R}, \mathbb{R}))$, whose family of activations only contains affine functions.

If $T$ is non-dense and non-trivial, by Lemma 12 we have two cases: if $\mathcal{F}(\mathcal{M})$ only contains affine functions we reduce to the maximal admissible pairs $(\mathcal{M} \subseteq \mathcal{R}_n(\mathbb{R}), \mathrm{Aff}(\mathbb{R}, \mathbb{R}))$ and $(\mathrm{GL}_n(\mathbb{R}), \mathrm{Hom}(\mathbb{R}, \mathbb{R}))$, otherwise, due to Lemma 12, $\mathcal{M} < \mathcal{M}_n(T)$ and $\mathcal{F}(\mathcal{M}) = \mathcal{F}_T$ by Lemma 11 and $\mathcal{M}(\mathcal{F}_T) = \mathcal{M}_n(T)$ because the inclusion $\mathcal{M}_n(T) \subseteq \mathcal{M}(\mathcal{F}_T)$ is obvious and if $\mathcal{M}(\mathcal{F}_T)$ contains non-monomial matrices, Lemma 12 would contradict $\mathcal{F}_T$ containing non-affine functions.

Applying Lemma 1, we get the admissible maximal pairs $(\mathcal{M}_n(T), \mathcal{F}_T)$. Finally, if $T$ is trivial, $\mathcal{M} = \mathcal{P}_n$ hence we obtain the maximal admissible pair $(\mathcal{P}_n, \mathcal{C}(\mathbb{R}))$ as verifying $\mathcal{F}(\mathcal{P}_n) = \mathcal{C}(\mathbb{R})$ is trivial and Lemma 12 implies $\mathcal{M}(\mathcal{C}(\mathbb{R})) = \mathcal{M}_n(\{1\}) = \mathcal{P}_n$. $\qquad\square$

We are now ready to prove Theorem 1.

*Proof.* Theorem 9 classifies admissible pairs whose relevant families of activations are $\mathcal{F}_T$ where $T$ varies on the multiplicative subgroups of $\mathbb{R}$. Thanks to the complete classification of $T$-multiplicative functions, $\mathcal{F}_T$, provided by Lemma 10, we can substitute them with their concrete counterparts enumerated in the statement. $\qquad\square$

In what follows we restate Theorem 2 and prove it.

**Theorem 10.** *Let $G$ be a compact group and let us restrict to families of activations containing some non-affine functions. The only two maximal admissible pairs up to isomorphism of groups of matrices are*

- *Continuous functions and permutation matrices,*

- *Odd continuous functions and signed permutation matrices.*

*Proof.* At first, we want to prove that each admissible representation of $G$ is isomorphic either to a permutation representation or to a sign-permutation representation. We split the proof into two parts. In the first part, $G$ is represented by non-negative monomial matrices and, in the second, $G$ is represented by arbitrary monomial matrices. Thanks to Theorem 1, those two cases covers all the admissible cases for families of activations containing at least one non-affine function.

By Flor (1969), each bounded non-negative group of matrices is isomorphic to a permutation group of matrices by a positive scaling of basis. Hence, the image of a compact group in $\mathcal{M}_n(T)$ with $T \subseteq \mathbb{R}_{>0}$ can be written as a group of permutation matrices after a positive scaling of basis. Hence, all the considered group of matrices are isomorphic to $\mathcal{P}_n$, therefore reducing to the pair $(\mathcal{P}_n, \mathcal{C}(\mathbb{R}))$.

Now consider $T$ an arbitrary non-dense multiplicative subgroup of $\mathbb{R}$. Each monomial matrix can be written as the product $SDP$, where $S$ is a diagonal matrix containing only $\pm 1$, $D$ a positive diagonal matrix, and $P$ is a permutation matrix. Consider the map $\phi : SDP \mapsto DP$. Note that $\phi$ is a continuous group homomorphism and that its image is a compact group of non-negative matrices. Indeed, $\phi$ is just the absolute value function defined on all the elements of the matrices, hence it is continuous. Then, let $S_1 D_1 P_\sigma$ and $S_2 D_2 P_\tau$ be two monomial matrices as before, where $P_\sigma$ and $P_\tau$ are permutation matrices respectively representing permutations $\sigma$ and $\tau$. Then their composition $S_1 D_1 P_\sigma S_2 D_2 P_\tau = S_1 \sigma(S_2) D_1 P_\sigma D_2 P_\tau$ where $\sigma(S_2)$ is the diagonal matrix obtained by permuting the diagonal elements through $\sigma$, i.e. $\sigma(S_2) = P_\sigma^t S_2 P_\sigma$, which commutes with $D_1$ being both diagonal matrices. This proves that $\phi$ is an homomorphism.

For what we proved at the beginning of the proof, there exists a positive scaling, represented by a positive diagonal matrix $B$, such that $BDPB^{-1} = P'$ is a permutation matrix for each $DP$ in the image of $\phi$. Note that for each $SDP$, after scaling by $B$, we obtain $BSDPB^{-1} = SBDPB^{-1} = SP'$ a signed permutation matrix. This is true because $S$ and $B$ commute being both diagonal matrices. Hence, all the considered groups of matrices are isomorphic to the group of signed permutation matrices, therefore, therefore reducing to the pair of odd continuous functions and signed permutation matrices.

Now that we have proven that there are only two isomorphism classes of admissible representations for $G$, we want to show that there is only one representation in each class with maximal family of activation functions. Let $\mathcal{M}$ be the image of a positive monomial matrices representation of $G$. By Lemma 11, each admissible group of matrices $\mathcal{M}'$ isomorphic to $\mathcal{M}$ commutes with $\mathcal{F}(\mathcal{M}') = \mathcal{F}_T$ for some $T$. The maximal family $\mathcal{F}_T$ is $\mathcal{C}(\mathbb{R})$ and it happens when $T = \langle 1 \rangle$ and $\mathcal{M}'$ are permutation matrices. Note that the isomorphism between $\mathcal{M}$ and some group of permutation matrices is always possible as shown at the beginning of this proof. An analogous argument applies to groups of arbitrary monomial matrices.

We conclude by noticing that in general permutation representations and signed-permutation representations are not isomorphic, e.g. the trivial representation and the sign representation of $S_n$. This proves that the two presented pairs are *in general* disjoint. $\square$

Note that compactness is a required condition. Indeed, given $b > 1$, consider the following one-dimensional representation representation $\rho : \mathbb{Z} \to \mathrm{GL}_1(\mathbb{R}) = \mathbb{R}^*$ such that $\rho(n)x = b^n x$. The image $\rho(\mathbb{Z})$ is not bounded and hence is not compact. This means that $\rho$ cannot be isomorphic to a permutation representation whose image is compact.

