# OpenReview forum: "A Characterization Theorem for Equivariant Networks with Point-wise Activations"
_ICLR.cc/2024/Conference — ICLR 2024 poster_

### Official Review · Reviewer_merJ · 2023-11-01

**Soundness:** 3 good
**Presentation:** 3 good
**Contribution:** 4 excellent
**Rating:** 8
**Confidence:** 3

**Summary:**

The paper studies the effect of pointwise activation on multilayer equivariant neural networks and establishes a characterization theorem of which activations can be used to construct a fully equivariant net for a wide variety of group and group action through the lens of representation theory. Notably, they give a unifying view for permutation-equivariant architectures, such as DeepSets and IGNs, which also contain some unexplored architectures; as well as showing that equivariant disentangled steerable CNN (in some definition) must be invariant if the activation is pointwise.

**Strengths:**

The paper highlights one key challenge in designing a fully equivariant neural network: namely the choice of activation function (common choices of which are not equivariant - e.g. commute with the linear part). Research into this topic is timely, highly relevant and important in the field.

Their framework generalizes existing work from Wood and Shawe-Taylor, and is tight in the sense that the characterization theorem enumerates all pairs of pointwise activation and linear layer that commute (and hence can be used in an equivariant network design).

The result for rotational equivariant architecture (that pointwise activation is equivariant iff representation of the linear layer is trivial) is surprising and may be crucial in opening up efforts to find other approaches to rotational equivariant architecture.

**Weaknesses:**

The paper is rather dense in representation theoretic notions. While I believe the paper is still self-contained, it may be beneficial for readers if the authors can expand on some intuition on the main theorems, maybe even in the appendix.

It doesn’t look like the technique in this paper can generalize to infinite dimensional representation, which the author acknowledged in Page 4.

Minor: I think the paper would also greatly benefit from a further discussion of "what's next?", what is the loss if we give up point-wise activations for rotational equivariant network but replacing them with other kinds of activation. For permutation-equivariant network, it may also be interesting to work out another nontrivial architecture that is neither DeepSets nor IGN (higher order tensor representation). I'd also understand if the authors deem these as being out-of-scope.

**Questions:**

Small typos:
- Page 15, proof of Lemma 4. First sentence in the proof misses “for all i”. Second sentence should start: “Then \sum_{j = 1}^n M_{ij} = 1” (missing “=1”).
- Page 15. First equation in Lemma 5 missing “i \in [n]” (missing “i”).
- Section 5.2. It is more apt to reference Cohen & Welling 2014 "Learning the irreducible representation..." rather than Cohen & Welling 2016b "Steerable CNN" in this section.

---

> ### Author Response · Authors · 2023-11-20
> **Response to Reviewer merJ**
>
> We thank the reviewer for the positive review, appreciation, and constructive feedback.
>
> **Response to weaknesses:**
>
> > The paper is rather dense in representation theoretic notions. While I believe the paper is still self-contained, it may be beneficial for readers if the authors can expand on some intuition on the main theorems, maybe even in the appendix.
>
> Regarding the first point, we are currently enhancing the paper's readability and comprehension for a broader audience. In particular, we will incorporate examples and strengthen the appendices.
>
> > It doesn’t look like the technique in this paper can generalize to infinite dimensional representation, which the author acknowledged in Page 4.
>
> Concerning the issue of infinite-dimensional representations: these play a significant role in the current theoretical discourse on equivariant models and our framework does not cover this approach. We acknowledge that this is one of the main limitations of our work, and we will address the matter in the manuscript with more precision. It is however remarkable to note that, while the descriptive models employed generally involve infinite-dimensional representations and concepts of harmonic analysis, the final deployed model is essentially a discretization of their theoretical formulation. In certain instances [1, 2], these models maintain exact equivariance solely through finite-dimensional representations and finite groups, conditions addressed in our framework as mentioned on page 4, though perhaps too briefly.
>
> > Minor: I think the paper would also greatly benefit from a further discussion of "what's next?", what is the loss if we give up point-wise activations for rotational equivariant network but replacing them with other kinds of activation.
>
> To address another problem raised by the reviewer, we will expand the conclusions. Despite the heterogeneous literature on non-pointwise activations, where each approach may vary for different tasks and little theoretical clarity exists, we will provide a more in-depth discussion.
>
> > For permutation-equivariant network, it may also be interesting to work out another nontrivial architecture that is neither DeepSets nor IGN (higher order tensor representation). I'd also understand if the authors deem these as being out-of-scope.
>
> Regarding the reviewer's last request, we can readily provide another example of a permutation-equivariant network that is not an IGN. For instance, considering the alternating group $A_n$ as the subgroup of even permutations in $S_n$, this subgroup cannot be isomorphic to any of the subgroups $S_\lambda$ presented in the paper. The module $\mathbb{R}^{S_n / A_n}$ will be a two-dimensional representation compatible with point-wise activations. Exploring such spaces, although they may seem pathological and of dubious practical utility, could aid in the understanding of permutation equivariance in its generality and hence in constructing better performing networks.
>
> **Response to questions:**
>
> We thank the reviewer again for the help in improving the manuscript. We acknowledge the errors and inaccuracies and want to assure that we are already incorporating the suggested corrections.
>
> **References:**
>
> [1] Taco S. Cohen and Max Welling, Group Equivariant Convolutional Networks. Proceedings of the 33 rd International Conference on Machine Learning, https://arxiv.org/pdf/1602.07576.pdf
>
> [2] Taco S. Cohen, Maurice Weiler, Berkay Kicanaoglu, Max Welling, Gauge Equivariant Convolutional Networks and the Icosahedral CNN, Proceedings of the 36 th International Conference on Machine Learning, https://arxiv.org/abs/1902.04615

---

> > ### Comment · Reviewer_merJ · 2023-11-21
> >
> > I'd like to thank the reviewer for their comment and confirm that I am keeping my score.

---

### Official Review · Reviewer_3Kou · 2023-11-03

**Soundness:** 3 good
**Presentation:** 2 fair
**Contribution:** 2 fair
**Rating:** 6
**Confidence:** 3

**Summary:**

The paper studies equivariance of neural network layers consisting of a linear transformation followed by a pointwise activation. The main result is a theorem characterizing largest admissible combinations of activation families and the associated groups for which equivariance holds. After that, the main result and its consequences are discussed in the context of several specific equivariance scenarios.

**Strengths:**

The main result - Theorem 1 - seems to be new (though heavily relying on previous research, primarily Wood & Shawe-Taylor (1996)). This theorem provides an explicit classification of equivariant pairs of activation families and groups and seems to be a simple, but useful result. I have not spotted any mistakes in the proofs.

**Weaknesses:**

**Contribution.** I doubt that the paper in its present state is strong enough for ICLR.
1. The main result looks like a relatively simple add-on to the study of equivariant networks by Wood & Shawe-Taylor (1996). Most results found in the appendix are either simple facts from the group and representation theory or are borrowed from Wood & Shawe-Taylor (1996) (e.g., the key equivariance condition (3)). Theorem 1 seems to be a relatively simple consequence of these results.

2. Though the paper includes a discussion of applications of Theorem 1 in Sections 5 and 6, I found these sections not easy to follow. They essentially consist of a long and not so well structured list of comments surrounding a few corollaries. The context of these comments is explained relatively vaguely, partly through references to other papers. It is not really clear to me why any of these corollaries and comments is actually important. The paper does not include specific examples, illustrations or experiments showing how Theorem 1 can, say, help improve designs of equivariant models.

**Writing.** The writing is sloppy in several places, with typos and unclear grammar. For example, only in Lemma 5:
- the index $i$ seems to be missing in the definition of $\mathcal{T}(\mathcal M)$;
- the indices and coefficients are wrong in the formula for $f(\sum_{j\in S}M_{rj}x)$;
- a word seems to be missing in "It is a simple to check".

I could understand the proof of Lemma 5 only with some effort (especially the sentence " And obviously each.."). The sentence "Hence, we can restrict our classification.." after the proof of Lemma 7 seems to consist of two different statements, but it is unclear where the first one ends and the second begins. In Lemma 7: "dived into three different type".

**Questions:**

I suggest to:
- carefully revise the writing;
- strengthen and clarify sections 5-6 by providing experiments and/or theorems so as to make it clear why the discussions there are important (it might be worth it to focus on fewer applications, but with more depth).

---

> ### Author Response · Authors · 2023-11-20
> **Response to Reviewer 3Kou**
>
> We thank the reviewer for the comprehensive assessment.
>
> **Response to contribution:**
>
> 1.
> > The main result looks like a relatively simple add-on...
>
> We agree with the reviewer, and explicitly acknowledge in the paper, that our proposed results are largely built on top of the pioneering work by Wood and Shawe-Taylor [1].
> However, we introduce the following extensions and improvements:
>
> * We generalize the theorem to non-finite groups and continuous activations, as new classes of networks emerge from this extension.
> * We note the importance of identifying certain network classes by considering representations up to isomorphism. Note that two isomorphic representations are equivalent in terms of learning capabilities and this aspect was not covered by the original paper. Indeed, we solve an issue raised in Section 4.2 of Wood and Shawe-Taylor where they provide an example of a positive monomial representation that is not a permutation representation. That representation is actually isomorphic to a permutation representation. With our approach we fix all issues like this in a *general* and *complete* setting.
> * We integrate the theory developed by Wood and Shawe-Taylor with several adjustments that make the classification effective. For example, our Lemma 1 ensures the maximality of the families of matrices and activations presented. This aspect is not discussed by Wood and Shawe-Taylor, and it is necessary for the rigor of the characterization.
> * We want to highlight that despite acknowledging the authorship of some technical lemmas presented at the end of the appendices to Wood and Shawe-Taylor, our proposed proofs place a greater emphasis on algebraic aspects, especially grounding our characterization on the classification of multiplicative subgroups of $\mathbb{R}$, diverging by Wood and Shawe-Taylor. This approach not only improves the readability of the proof but also highlights the potential to generalize these results to complex numbers as a plausible solution to certain limitations of real equivariant networks as suggested in the Conclusions.
>
> In the manuscript that we will upload, we will further update and refine these proofs. When compared to the original ones, they will show their kinship, but also their novelty.
> We believe that the impact and implications of these innovations are significant for the machine learning community, especially considering the evolution and growth of the use and study of equivariant representation learning during the twenty years passed since the publication of Wood and Shawe-Taylor's paper.
>
> 2.
> > Though the paper includes a discussion...
>
> We thank the reviewer for pointing out the difficulty in understanding the focus and significance of Sections 5 and 6. We intend to refine these sections in the updated version of the manuscript to enhance comprehension. This will involve introducing fundamental concepts essential for understanding, along with the inclusion of simple examples. In instances where space constraints may limit these improvements, we will ensure to adequately strengthen the appendices and incorporate appropriate references in the main text. The reviewer rightfully questions how our proposed results may influence the design of equivariant models. We indirectly address this concern by highlighting how our findings show gaps in understanding specific phenomena. For instance, in Section 5.2, we present a list of permutation-equivariant neural networks which, to the best of our knowledge, are not found in the existing literature. The existence of these networks show an inadequate understanding of the permutation equivariance, a gap stemming from the scarcity of theoretical results in the field. In the long term, addressing this gap appropriately may lead to the development of new, more efficient, and better-performing models. With this work, we aim to contribute to this endeavor.
>
> **Response to writing:**
>
> We thank the reviewer for their comments, particularly appreciating the precision of their remarks. We acknowledge the errors identified in Lemma 5 and will promptly rectify them in the manuscript. We will also revise the proof of Lemma 1 to enhance its clarity. Furthermore, we acknowledge and will rectify the errors and ambiguities noted in Lemma 7.
>
> **Response to questions:**
>
> As mentioned earlier, we are committed to enhancing the readability of the paper by improving the quality of writing and incorporating illustrative examples. Regarding the suggestion to focus on fewer applications but delve into them more thoroughly, we acknowledge this as a viable and effective approach. Secondly, maintaining a comprehensive overview of all applications is essential to convey the significance and ubiquity of the implications of Theorem 1 within the entire research field of geometric deep learning.
>
> **References:**
> [1] Wood and Shawe-Taylor. Representation theory and invariant neural networks https://www.sciencedirect.com/science/article/pii/0166218X95000753

---

> > ### Comment · Reviewer_3Kou · 2023-11-23
> >
> > I thank the authors for their clarifications and addressing my concerns. I am raising my score.

---

### Official Review · Reviewer_uXzC · 2023-11-04

**Soundness:** 3 good
**Presentation:** 1 poor
**Contribution:** 2 fair
**Rating:** 6
**Confidence:** 2

**Summary:**

This works considers equivariant neural networks, consisting of equivariant affine maps and pointwise nonlinearities that map between representations of a group $G$. They provide a characterization theorem, which simultaneously characterizes possible types of pointwise nonlinearities and maximal group representations that commute with the families of pointwise nonlinearities. This theorem is applied to several specific types of group representations that have been studied in the equivariant machine learning literature.

**Strengths:**

1. Substantially generalizes previous existing results, of which there are not many.
2. The main result is strong, and remarkably shows that there are not too many pairs of maximal admissible families of activations and groups.
3. The argument on page 7 for why it is better to use pointwise activations in the usual representation for $S_n$ on $\mathbb{R}^n$ as opposed to a disentangled representation is nice. The equivariant linear maps and pointwise activation approach is universal with the usual representation due to Segol and Lipman 2019 (https://arxiv.org/abs/1910.02421), but can be very weak in a disentangled representation.

**Weaknesses:**

1. I find this paper quite difficult to read at times, and I think that it would benefit from more explicitly defining certain concepts and writing out certain arguments. In the proof of Corollary 3, I do not quite understand the application of Theorem 1; why does the representation have to be trivial? Also, Section 6.2 about invariant graph networks is very dense. See also questions.
2. The applicability of these results to empirical equivariant machine learning is unclear. Already practitioners do not use pointwise nonlinearities for $SO(3)$ equivariant networks. Also, are there any expectations of potential benefits from using other representations of $S_n$ for invariant graph networks?

Overall, I have trouble reading the paper in its current state, but I am very open to changing my assessment of this paper in the discussion stage.

**Questions:**

1. In Section 6.1 / Section 6.3, why is the following case ruled out? Say I let $f$ be linear, so that $\tilde f$ is just scaling and is hence equivariant to $SO(3)$. Then the output representation is the same as the input, and is in particular not invariant. This can be done in the $SO(3) \times S_n$ case as well.

Other notes:
1. Typo: Page 4, sentence with "Although employing techniques analogue to..." has several typos that make it confusing, incluidng use of "hypothesis" and "thesis"
2. Typo: Page 14, "Aff" is incorrectly typeset before Theorem 5

---

> ### Author Response · Authors · 2023-11-20
> **Response to Reviewer uXzC**
>
> We thank the reviewer for the thorough comments.
>
> **Response to weaknesses:**
>
> **Readability:**
> > 1. I find this paper quite difficult to read at times, and I think that it would benefit from more explicitly defining certain concepts and writing out certain arguments. In the proof of Corollary 3, I do not quite understand the application of Theorem 1; why does the representation have to be trivial? Also, Section 6.2 about invariant graph networks is very dense. See also questions.
>
> As suggested, we will provide more explicit definitions of concepts and arguments. In particular, we will modify the proof of Corollary 3 to emphasize how it derives from Theorem 1 and how those representations are trivial. Subsequently, we will proceed to clarify and expand Section 6.2.
>
> **Applicability to equivariant ML:**
> > 2. The applicability of these results to empirical equivariant machine learning is unclear.
>
> This theorem may be of interest even to the ordinary machine learning practitioner as it provides the complete list of cases where the application of point-wise activations is viable and those can generally lead to increased computational efficiency.
>
> **Result on SO(3):**
> > Already practitioners do not use pointwise nonlinearities for $SO(3)$-equivariant networks.
>
> We agree that many different approaches to non-pointwise activations for equivariant $SO(3)$-equivariant networks are employed in practice but they in general require more computational resources, present a slow convergence rate, or both. For this reason, it may be favorable to employ pointwise activation to mitigate those shortcomings, but we prove in the paper that this is not a viable option.
> Indeed, we believe that this result is of greater significance for machine learning researchers aiming to construct new equivariant models. In fact, in some relevant cases (e.g. exact rotation equivariance), this theorem suggests that natural research directions (e.g., pointwise activations coupled with any $SO(3)$-representation) might require more careful thinking.
>
> **Implication for IGN:**
> > Also, are there any expectations of potential benefits from using other representations of $S_n$ for invariant graph networks?
>
> Regarding the list of other presented $S_n$-equivariant models, we have no strong results about their general properties. We deemed it significant to highlight their existence not only for sake of completeness but also because this manifests a lack of understanding of the broader concept of permutation equivariance, a deficiency arising from a scarcity of theoretical work in the field, a lacuna that, if strongly addressed, could lead to the construction of more efficient and expressive models. With this work, we hope to contribute positively to the long-term effort of filling this gap.
>
> **Response to questions:**
> > 1. In Section 6.1 / Section 6.3, why is the following case ruled out? Say I let $f$ be linear, so that
> $\tilde f$ is just scaling and is hence equivariant to $SO(3)$. Then the output representation is the same as the input, and is in particular not invariant. This can be done in the $SO(3)
> \times S_n$ case as well.
>
> We thank the reviewer for catching our oversight formulating Corollary 3; indeed, an hypothesis was missing from the statement and the statement holds true only for nonlinear activations which rules out the provided example.
>
> **Typos:**
>
> We appreciate the reviewer for pointing out these typos. The corrections will be promptly incorporated into the manuscript.

---

> > ### Comment · Reviewer_uXzC · 2023-11-20
> >
> > I thank the authors for their clarifications, and promises to improve the exposition of the paper. I will raise my score from a 5 to a 6.

---

### Official Review · Reviewer_sg92 · 2023-11-06

**Soundness:** 4 excellent
**Presentation:** 2 fair
**Contribution:** 3 good
**Rating:** 8
**Confidence:** 3

**Summary:**

The paper describes a theorem to characterize what group representations can be used given certain types of point-wise activation functions, and vice versa. A main result is that either permutation representations can be used, or trivial irreducible representations. The paper discusses several equivariant networks in context of the obtained results.

[update: based on the rebuttal I raised my score]

**Strengths:**

1. The paper presents a solid contribution to understanding the limits of equivariant neural networks in light of point-wise activation functions
2. The paper is thorough and precise in its definitions and results (theorems and proofs)
3. The paper is open about it's scope (I appreciate the comment above Section 4 about not addressing homogeneous spaces / infinite dimensional vector spaces) and defines promising directions for future research.

**Weaknesses:**

1. The paper is a theoretical contribution and emphasizes that the results are of practical value. However, I believe that in order for it to be of practical value the paper should be more accessible to a broader audience. Although formal and precise, there is little effort put in describing the main results in plain English.
2. Although partially covered by clarifying the scope of the paper, I do think the notion of regular representations is insufficiently covered. By sticking to irreducible representations one naturally comes to the conclusion that if one wants to apply point-wise activations one has to use trivial (invariant) irreps. However, if one admits regular representations (either approximations in the full SO(n) case, or exact as in the discrete sub-group cases of SO(n)) then one is not limited to invariants as the regular representations are effectively permutation representations. The regular representations can also be defined in the continuous setting, though the discretization may introduce some equivariance error, however, this need not be a problem as one could use random grids to avoid such biases. See e.g. Kuipers-Bekkers-2023:

Kuipers, T.P., Bekkers, E.J. (2023). Regular SE(3) Group Convolutions for Volumetric Medical Image Analysis. In: Medical Image Computing and Computer Assisted Intervention – MICCAI 2023. LNCS, vol 14222. Springer, Cham. https://doi.org/10.1007/978-3-031-43898-1_25
https://arxiv.org/abs/2306.13960

As for equivariant models with point-wise activation functions, it is often the case that internally the models use (non-trivial) irreps to parametrize the layers, and use an inverse Fourier transform to transform back to the regular representation basis, apply point-wise activation, then project back to irrep basis. I think including such approach in the discussions enable more people to relate to the result of the paper. Although commonly used in practice, I did find it hard to find references to such approaches. See e.g. the library e3nn ( https://docs.e3nn.org/en/latest/api/nn/nn_s2act.html ) and escnn (https://quva-lab.github.io/escnn/api/escnn.nn.html?highlight=fourier#escnn.nn.FourierPointwise). As for papers I could find [Cesa et al. 2021] Appendix H.2:

Cesa, G., Lang, L., & Weiler, M. (2021, October). A program to build E (N)-equivariant steerable CNNs. In International Conference on Learning Representations.

and perhaps the following as it represents signals on a grid but performs convs in the Fourier domain

Cohen, T. S., Geiger, M., Köhler, J., & Welling, M. (2018, February). Spherical CNNs. In International Conference on Learning Representations.

3. By the previous item, remarks such as (see abstract) "we prove that rotation-equivariant networks can only be invariant" is in practice more nuanced, and I consider the statement actually incorrect. Another example of a rotation equivariant method based on equivariantly obtained spherical grids (they call it equivariant mesh) is GemNet:

Gasteiger, J., Becker, F., & Günnemann, S. (2021). Gemnet: Universal directional graph neural networks for molecules. Advances in Neural Information Processing Systems, 34, 6790-6802.

Effectively, the paper works with scalar fields over the homogeneous space $\mathbb{R}^3 \times S^2$, as also discussed in recent work https://arxiv.org/abs/2310.02970 , which also fully operates in the domain of regular representations. I remark that this reference seem to have appeared only after the ICLR deadline so of course I do not expect this to have been included.

4. In conclusion, I think the paper could improve on impact if the relation to regular group convolution methods is further clarified and if the main results are introduced in simpler terms. The claim that equivariant networks can only be invariant is I think an artifact of the choice of sticking to irreducible representations.

Thank you for the great work and for considering my comments. I'm open to discussing my comments as I could have surely misinterpreted some parts of the paper.

**Questions:**

See above comments and smaller details below:

1. On page 3 the symbol $\mathbb{R}^*$ is used. What does the asterix indicate?
2. I did not understand the introduction of the main result "Although employing techniques analogue to already known ones, we highlight how hypothesis can generalize ... of the basis" Apart from this sentence being too long, I really do not understand what is being said here, please rewrite.
3. Theorem 1 could benefit from some subsequent, explicit examples. Take ReLU and e.g. a cyclic permutation group. Or any example you see fit. I think the paper remains abstract throughout and I think it could be helpful to see some common examples.
4. Section 5.1 is also a bit abstract still, dispite being in the section for practical scenarios. Some laymans explainations of the quotient structure could be helpful. In particular I struggled with "...an equivariant isomorphism between ... which is the standard representation space for permutations equivariant networks on sets".
5. Section 5.2 makes sense, but as discussed it is limited in scope regarding the use of irreducible representations exclusively.
6. Could you clarify Corrolary 3. It is hard to understand what is about. In particular what is the significance of the connected component (are we talking about subgroups here?)
7. "... but not more general equivariant tasks such as segmentation or detection" again this is an artifact of sticking to irreps.

---

> ### Author Response · Authors · 2023-11-20
> **Answer to Weaknesses**
>
> We thank the reviewer for the valuable comments and suggestions.
>
> In the following, we present responses to the comments organized in a manner analogous to the structure of the review.
>
> 1. **Readability and accessibility:**
>
> > 1. The paper is a theoretical contribution and emphasizes that the results are of practical value. [...]
>
> We acknowledge the importance of improving the paper's accessibility and readability. In our revisions, we ensure to present the main results in a more straightforward manner to enhance clarity for a broader audience.
>
> 2. **Regular representations and Irreps:**
>
> > 2. Although partially covered by clarifying the scope of the paper, I do think the notion of regular representations is insufficiently covered. [...]
>
> If we correctly understand the concerns regarding regular representations and irreps, we suppose that there may be confusion due to the adoption of Wood and Taylor's notation for regular representations. We will explicitly clarify in the manuscript that despite this notation choice, our results address arbitrary finite-dimensional representations, which encompass regular and, more in general, permutation representations. Additionally, we will emphasize that the specific treatment of disentangled representations (hence the introduction of irreps) is confined to Section 5.2 and does not affect other parts of the paper. The invariance result for compact connected topological groups (e.g. $SO(3)$) is not implied by disentanglement but it follows from the finiteness of the matrix groups underlying admissible representations. Indeed, note that hypotheses of Corollary 3 do not require disentanglement. For further clarification, the last paragraph in Section 6.3 exploits the existence of decomposition in irreducible components and Schur's Lemma only to prove triviality, but it does not require disentanglement of the layer, meaning that the basis chosen in the definition of the pointwise activation (see Section 3.2) does not need to agree with some irrep decomposition (see the first paragraph of Section 5.2). Note that irrep decompositions exist for each representation, regular ones too, (see Appendix A.2, Theorem 2) the stringent constraint is disentanglement that we are not assuming in this instance.
>
> 3. **Discretization:**
>
> > 3. By the previous item, remarks such as (see abstract) "we prove that rotation-equivariant networks can only be invariant" is in practice more nuanced, and I consider the statement actually incorrect. [...]
>
> In agreement with the reviewer, we acknowledge the necessity for a broader contextualization of Theorem 1 within the current literature covering all nuanced takes on equivariance. For brevity in this comment, we categorize these nuances as approximate equivariance compared to exact equivariance, and finite-dimensional frameworks compared to harmonic analysis approaches. Adopting this simplified view, our restriction to  exact equivariance and finite-dimensional representations should be pointed out earlier and more clearly in the text, highlighting how the analysis conducted under these constraints also affects practices derived from harmonic analysis approaches. We proceed to elaborate on the matter in greater detail:
> We agree, as pointed out by the reviewer, that a substantial portion of the current literature on equivariant models relies on harmonic analysis approaches [1, 2, 4]. Processed inputs are continuous finite-energy signals defined on homogeneous spaces (e.g., $L^2(G)$ functions) which are infinite-dimensional spaces and thus fall outside the scope of our study.
> However, as pointed out at page 4 of the paper, in practice continuous models are often discretized, relaxing the equivariance condition to an approximate notion. This process is often obtained by truncating finite Fourier transforms of the signal or discretizing the symmetry group and domain. This discretization frequently boils down to considering equivariant models with finite-dimensional representations of finite groups [5, 8], which is dealt with by our framework and, in line with our findings, allow pointwise activations and non-trivial representations.
> Contrary to our approach, other methods suggested by the reviewer (such as GemNet [3]) employ tools akin to Tensor Field Networks [7], featuring non-pointwise activations that, as implied by the paper's title, go beyond the scope of our work (utilizing tensor representations, a nonlinear function, formula 4 in Dym and Maron [7]).
> These are however very interesting directions that we will address as future research.
>
> 4. **Conclusion:**
>
> > 4. In conclusion, I think the paper could improve on impact if the relation to regular group convolution methods is further clarified and if the main results are introduced in simpler terms. [...]
>
> We agree, and in light of the observations made, we will incorporate your suggestions and compare our work to the proposed references to enhance the overall impact and clarity of the paper.

---

> > ### Comment · Reviewer_sg92 · 2023-11-22
> >
> > Thank you for the clarification! I think the paper is of good overall quality and hence recommend accept (6). The paper is sound and overall accessible.
> >
> > My worry about the claim of practical usefulness remains. I am afraid that stating things like "we prove that rotation-equivariant networks can only be invariant" will only hamper practical usefulness (/impact) as in no one would be interested in neural networks that are strictly invariant. I very much appreciate our discussion above, however I did not see how this discussion will be incorporated in the revision, and thus cannot justify an increase in my score and maintain it at 6.
> >
> > Finally a small note, GemNet [3] does use point-wise activation functions. It is based on invariant message passing, but effectively on the homogeneous space of positions and orientations (as messages are send between edges), see [4]. The precise group convolutional viewpoint in that paper is obscured, but it is an instance of an exactly equivariant neural network, which utilizes point-wise non-linearities. It achieves exact equivariance through the construction of "equivariant meshes". The paper is of a different kind than TFNs (I understand the confusion, but only uses part of the theory for proving universal approximation power [7], and only uses spherical harmonics for embedding the invariants, it is not based on clebsch-gordan tensor products otherwise).
> >
> > Thus, the claim of invariance is technically sound from the disentangled representation (steerable/irrep) viewpoint, but I think the nuance and context as raised in our discussions are very important.

---

> ### Author Response · Authors · 2023-11-20
> **References and Answer to Questions**
>
> In the following, we present the responses to the questions posed in the second part of the review.
>
> 1. > On page 3 the symbol $\mathbb{R}^*$ is used. What does the asterix indicate?
>
> These are the multiplicatively invertible elements in $\mathbb{R}$. The definition is initially presented in Lemma 6 in the Appendix. We appreciate the reviewer for bringing this to our attention; we will introduce the definition in the main body of the text before referencing this set.
>
> 2. > I did not understand the introduction of the main result "Although employing techniques analogue to already known ones, we highlight how hypothesis can generalize ... of the basis" Apart from this sentence being too long, I really do not understand what is being said here, please rewrite.
>
> This segment highlights how our work contributes to advancing the state of the art, specifically building upon the work of Wood and Shawe-Taylor [6]. Notably, we want to emphasize how we achieve more refined results by considering representations up to isomorphism, despite employing more general assumptions. We will ensure to enhance the presentation of this section in the manuscript.
>
> 3. > Theorem 1 could benefit from some subsequent, explicit examples. Take ReLU and e.g. a cyclic permutation group. Or any example you see fit. I think the paper remains abstract throughout and I think it could be helpful to see some common examples.
>
> We appreciate feedback on Theorem 1 and agree that providing explicit examples, such as employing ReLU with a cyclic permutation group, would enhance the clarity and applicability of the paper. We will certainly integrate these examples into the paper to offer a more concrete understanding of the theoretical concepts presented.
>
> 4. > Section 5.1 is also a bit abstract still, dispite being in the section for practical scenarios. Some laymans explainations of the quotient structure could be helpful. In particular I struggled with "...an equivariant isomorphism between ... which is the standard representation space for permutations equivariant networks on sets".
>
> We will provide detailed explanations and examples of quotient groups and spaces induced by those groups. Although I believe we may need to move these additional elements to the appendices due to manuscript length constraints.
>
> 5. > Section 5.2 makes sense, but as discussed it is limited in scope regarding the use of irreducible representations exclusively.
>
> We concur. This section introduces a standalone corollary that does not impact the subsequent results we present. Nevertheless, given the role of disentangled networks in the literature, we deem it relevant to include this presentation.
>
> 6. > Could you clarify Corrolary 3. It is hard to understand what is about. In particular what is the significance of the connected component (are we talking about subgroups here?)
>
> In this scenario topological groups, such as $SO(3)$, come into play. These groups exhibit topological properties such as connectedness and compactness, which we employ in Corollary 3. We will ensure to provide additional details and examples in both the main text and the appendices.
>
> 7. > "... but not more general equivariant tasks such as segmentation or detection" again this is an artifact of sticking to irreps.
>
> This point stems from the finiteness of the matrix group induced by the representations. This statement is general and not specific to disentangled networks alone. We believe that our misalignment on this point can be traced back to the discussion in Answer 3 Discretization.
>
> **References:**
>
> [1] Kuipers, T.P., Bekkers, E.J. (2023). Regular SE(3) Group Convolutions for Volumetric Medical Image Analysis. https://arxiv.org/abs/2306.13960
>
> [2] Cohen, T. S., Geiger, M., Köhler, J., & Welling, M. (2018). Spherical CNNs.
>
> [3] Gasteiger, J., Becker, F., & Günnemann, S. (2021). Gemnet: Universal directional graph neural networks for molecules.
>
> [4] Fast, Expressive SE(n) Equivariant Networks through Weight-Sharing in Position-Orientation Space, Preprint, https://arxiv.org/abs/2310.02970
>
> [5] Taco S. Cohen and Max Welling, Group Equivariant Convolutional Networks, https://arxiv.org/pdf/1602.07576.pdf
>
> [6] Jeffrey Wood and John Shawe-Taylor. Representation theory and invariant neural networks. https://www.sciencedirect.com/science/article/pii/0166218X95000753
>
> [7] Nadav Dym and Haggai Maron. On the Universality of Rotation Equivariant Point Cloud Networks.
>
> [8] Taco S. Cohen, Maurice Weiler, Berkay Kicanaoglu, Max Welling, Gauge Equivariant Convolutional Networks and the Icosahedral CNN, https://arxiv.org/abs/1902.04615

---

### Author Response · Authors · 2023-11-20
**Response overview**

We thank all the reviewers for their comments and suggestions, which we believe will significantly enhance the manuscript's quality.

We are glad to see that the reviewers found:

1. Research into this topic timely, highly relevant and important in the field (Reviewer merJ).
2. Our work constitutes a significant contribution to understanding the design challenges of equivariant neural networks with point-wise activations. It has a wide scope, but we openly acknowledge its limitations (Reviewer sg92).
3. The methodology employed is mathematically precise and rigorous, as the reviewers have not identified any mistakes in the proofs (Reviewers 3Kou, sg92).
4. The classification is a simple yet useful result (Reviewer 3Kou), tight and complete. In other words, the theorem enumerates all pairs of pointwise activation and linear layers that commute (Reviewers merJ, uXzC).
5. The results represent a substantial generalization of the few previous findings (Reviewer uXzC).

In the following section, we offer a succinct overview of our responses addressing the most common concerns raised by the reviewers.

**Presentation and writing:** All reviewers highlight the readability and accessibility issues of the manuscript. We acknowledge this problem and commit to addressing it by revising the writing, incorporating additional descriptions, providing less formal clarifications and examples. Concurrently, we will enhance the appendices with fundamental theoretical details to make the work more self-contained.

**Limitations regarding infinite-dimensional representations and harmonic analysis:** Infinite-dimensional representations and harmonic analysis approaches hold a significant role in the ongoing theoretical discussions concerning equivariant models, an aspect not covered by our current framework. However, we argue that, despite the frequent involvement of infinite-dimensional representations in descriptive models, the ultimate computational model essentially is a discretization of a continuous formulation. In a considerable number of cases, these models maintain exact equivariance exclusively through finite-dimensional representations and finite groups, conditions explicitly addressed in our framework. Nevertheless, we do not intend to diminish the significance of research directions that fall outside the boundaries of our framework.

**Practical impact:** This theorem might capture the attention of not only researchers but also regular machine learning practitioners, as it offers the complete list of representations where the use of point-wise activations is feasible, potentially resulting in enhanced computational efficiency. Some reviewers appropriately question the potential impact of our proposed results on the design of equivariant models. We indirectly respond to this inquiry by reinforcing how our findings reveal deficiencies in understanding equivariant neural networks in their general form. A gap arising from the scarcity of theoretical results in the field. Over the long term, appropriately addressing this gap may result in the creation of new and efficient models. Our aim is to make a meaningful contribution along this direction.

As a final note, we wish to emphasize our commitment to providing coherent and detailed responses to each reviewer's comments. In case of unmet expectations, we strongly encourage reviewers to seek additional clarification. We remain open and available for discussions on any aspects that could improve the manuscript.

---

> ### Author Response · Authors · 2023-11-23
> **Manuscript Update**
>
> We have uploaded a new version of the draft, aiming to incorporate the changes suggested by the reviewers.
> First and foremost, our effort has been directed towards enhancing the overall accessibility of the work. We provided a clearer description of the results in plain English, incorporating examples, expanding the appendices with the requested introductory elements, and relocating more technical results to the appendices.
>
> In order to enhance text comprehensibility, we opted to split the statement of the main theorem into two theorems. The first theorem is the generalization of the work by Wood and Shawe-Taylor to continuous activation functions and arbitrary groups. The second theorem refines the outcome of the preceding theorem, up to isomorphisms of representations.
>
> Secondly, we improved the readability of the proofs of the theorems in the appendices. It should now be more evident what our contribution is in comparison to Shawe-Taylor's work. We have divided the proofs into multiple lemmas, aiming to achieve results that are simpler to comprehend, and striving to explain each aspect thoroughly. As elucidated in the text, the results we propose from Shawe-Taylor are only Lemma 11 and Lemma 12, while we believe the remainder of the results between pages 17 and 21 to be original.

---

> ### Comment · Reviewer_sg92 · 2023-11-23
>
> Thank you, I think the extra context and detail provided in the current manuscript are valuable additions. Based on it, I decided to raise my score.

---

### Meta-Review · Area_Chair_JBbX · 2023-12-06

**Metareview:**

This paper investigates the equivariant property of neural networks with point-wise activation functions. For the important problem that point-wise activation functions do not usually have the equivariant property, this paper uses representation theory to derive new properties for the associated functions and matrices. The results are both mathematically solid and new. Although some aspects of the practical impact are not yet clear enough, the importance of the problem and the novelty of the results are well worth the effort.

**Justification For Why Not Higher Score:**

The reviewers raised concerns about the practical merits of this paper.

**Justification For Why Not Lower Score:**

This paper introduces a new mathematical analysis of an important problem. Also, this paper is carefully written.

---

### Decision · Program_Chairs · 2024-01-16

Accept (poster)